# Applications of Common Entropy for Causal Inference

**Murat Kocaoglu**
MIT-IBM Watson AI Lab, IBM Research
murat@ibm.com

**Sanjay Shakkottai**
The University of Texas at Austin
shakkott@austin.utexas.edu

**Alexandros G. Dimakis**
The University of Texas at Austin
dimakis@austin.utexas.edu

**Constantine Caramanis**
The University of Texas at Austin
constantine@utexas.edu

**Sriram Vishwanath**
The University of Texas at Austin
sriram@austin.utexas.edu

## Abstract

We study the problem of discovering the simplest latent variable that can make two observed discrete variables conditionally independent. The minimum entropy required for such a latent is known as common entropy in information theory. We extend this notion to Rényi common entropy by minimizing the Rényi entropy of the latent variable. To efficiently compute common entropy, we propose an iterative algorithm that can be used to discover the trade-off between the entropy of the latent variable and the conditional mutual information of the observed variables. We show two applications of common entropy in causal inference: First, under the assumption that there are no low-entropy mediators, it can be used to distinguish causation from spurious correlation among almost all joint distributions on simple causal graphs with two observed variables. Second, common entropy can be used to improve constraint-based methods such as PC or FCI algorithms in the small-sample regime, where these methods are known to struggle. We propose a modification to these constraint-based methods to assess if a separating set found by these algorithms are valid using common entropy. We finally evaluate our algorithms on synthetic and real data to establish their performance.

## 1 Introduction

Understanding the causal workings of a system from data is essential in many fields of science and engineering. Recently, there has been increasing interest in causal inference in the machine learning (ML) community. While most of ML has traditionally been relying solely on correlations in the data, it is now widely accepted that distinguishing causation from correlation is useful even for simple predictive tasks. This is because causal relations are more robust to the changes in the dataset and can help with generalization, while an ML system relying solely on correlations might suffer when these correlations change in the environment the system is deployed in [12].

A causal graph is a directed acyclic graph that depicts the causal workings of the system under study [38]. Since it indicates the causes of each variable, it can be seen as a qualitative summary of the underlying mechanisms. Learning the causal graph is the first step for most of the causal inference tasks, since inference algorithms rely on the causal structure. Causal graphs can be learned from

randomized experiments [17, 14, 44, 28, 27]. In settings where performing experiments are costly or infeasible, one needs to resort to observational methods, i.e., make best use of observational data, potentially under assumptions about the data generating mechanisms.

There is a rich literature on learning causal graphs from observational data [47, 54, 11, 1, 35, 46, 20, 50, 9]. *Score-based* methods optimize a regularized likelihood function in order to discover the causal graph. Under certain assumptions, these methods are consistent; they obtain a causal graph that is in the correct equivalence class for the given data [34, 10]. However, score-based methods are applicable only in the causally sufficient setting, i.e., when there are no latent confounders. A variable is called a latent confounder if it is not observable and causes at least two observed nodes. *Constraint-based* methods directly recover the equivalence class in the form of a mixed graph [1, 34, 47, 35, 55]: They test the conditional independence (CI) constraints in the data and use them to infer as much as possible about the causal graph. Despite being well-established for graphs with or without latents, constraint-based methods are known to work well only with an abundance of data. Early errors in CI statements might lead to drastically different graphs due to the sequential nature of these algorithms. A third class of algorithms can be described as those imposing assumptions in order to identify the graphs which are otherwise not identifiable [20, 46, 39, 21, 26]. Most of this literature focuses on the cornerstone case of two variables $X, Y$ where constraint and score-based approaches are unable to identify if $X$ causes $Y$ or $Y$ causes $X$, simply because they are indistinguishable without additional assumptions. This literature contains a wide range of assumptions that we summarize in Section 7.2.

Information theory has been shown to provide tools that can be useful for causal discovery [8, 42, 30, 15, 52, 26, 48]. In this work, we explore the uses of *common entropy* for learning causal graphs from observational data. To define common entropy, first consider the following problem: Given the joint probability distribution of two discrete random variables $X, Y$, we want to construct a third random variable $Z$ such that $X$ and $Y$ are independent conditioned on $Z$. Without any constraints this can be trivially achieved: Simply picking $Z = X$ or $Z = Y$ ensures that $X \perp\!\!\!\perp Y \,|\, Z$. However, this trivial solution requires $Z$ to be as *complex* as $X$ or $Y$. We then ask the following question: *is there a simple $Z$ that makes $X, Y$ conditionally independent?* In this work, we use Rényi entropy of the variable as a notion of its complexity. Then the problem becomes identifying $Z$ with the smallest Rényi entropy such that $X \perp\!\!\!\perp Y \,|\, Z$. Shannon entropy of this $Z$ is called the *common entropy* of $X$ and $Y$ [31].

We demonstrate two uses of common entropy for causal discovery. The first is in the setting of two observed variables. Suppose we observe two correlated variables $X, Y$. Figure 1 shows some causal graphs that can induce correlation between $X, Y$. Note that *Latent Graph* differs from the others in that $X$ does not have any causal effect on $Y$. Then distinguishing the latent graph from the others is important to understand whether an intervention on one of the variables will cause a change in the other. We show that if the latent confounder is *simple*, i.e., has small entropy then one can distinguish the latent graph from the triangle and direct graphs using common entropy. To identify the latent graph, we assume that the correlation is not induced only by a simple mediator, which eliminates the mediator graph. We show that this is a realistic assumption using simulated and real data.

Second, we show that common entropy can be used to improve constraint-based causal discovery algorithms in the small sample regime. For such algorithms, correctly identifying *separating sets*, i.e., sets of variables that can make each pair conditionally independent, is crucial. Our key observation is that, for a given pair of variables common entropy provides us with an information-theoretic lower bound on the entropy of any separating set. Therefore, it can be used to reject incorrect separating sets. We present our modification on the PC algorithm, which is called the *EntropicPC* algorithm.

To the best of our knowledge, the only result for finding common entropy is given in [31], where they identify its analytical expression for binary variables. They also note that the problem is difficult in the general case. To address this gap, in Section 2 we propose an iterative algorithm to approximate common entropy. We also generalize the notion of common entropy to *Rényi common entropy*.

Our contributions can be summarized as follows:

- In Section 2, we introduce the notion of Rényi common entropy. We propose a practical algorithm for finding common entropy and prove certain guarantees. Readers interested only in the applications of common entropy to causal inference can skip this section.

- In Section 3, under certain assumptions, we show that $\text{Rényi}_0$ common entropy can be used to distinguish latent graph from the triangle and direct graphs in Figure 1. We also show this identifiability result via $\text{Rényi}_1$ common entropy for binary variables, and propose a

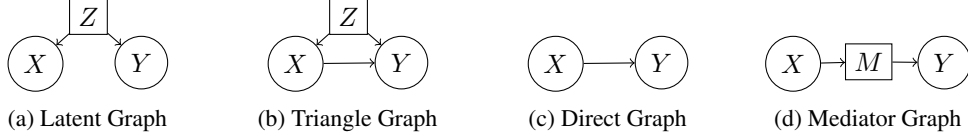

<center>(a) Latent Graph      (b) Triangle Graph      (c) Direct Graph      (d) Mediator Graph</center>

Figure 1: Different graphs that explain correlation between the observed $X, Y$. $Z, M$ are unobserved.

conjecture for the general case. In Section 5.2, we validate one of our key assumptions in real and synthetic data. In Section 5.3, we validate our conjecture via real and synthetic data.

- In Section 4, we propose *EntropicPC*, a modified version of the PC algorithm that uses common entropy to improve sample efficiency. In Section 5.5, we demonstrate significant performance improvement for EntropicPC compared to the baseline PC algorithm. We also illustrate that EntropicPC discovers some of the edges missed by PC in ADULT data [13].

- In Section 5, in addition to the above, we provide experiments on the performance of our algorithm for finding common entropy, as well as its performance on distinguishing the latent graph from the triangle graph on synthetic data.

**Notation:** Support of a discrete random variable $X$ is shown as $\mathcal{X}$. $p(.)$ and $q(.)$ are reserved for discrete probability distribution functions (pmfs). $[n] \coloneqq \{1, 2, \dots, n\}$. $p(Y|x)$ is shorthand for the conditional distribution of $Y$ given $X = x$. Shannon entropy, or entropy in short, is $H(X) = -\sum_x p(x) \log(p(x))$. Rényi entropy of order $r$ is $H_r(X) = \frac{1}{1-r} \log\left(\sum_x p^r(x)\right)$. It can be shown that Rényi entropy of order 1 is identical to Shannon entropy. $D = (\mathcal{V}, \mathcal{E})$ is a directed acyclic graph with vertex set $\mathcal{V}$ and edge set $\mathcal{E} \subset \mathcal{V} \times \mathcal{V}$. $Pa_i$ represents the set of parents of vertex $X_i$ in the graph and $pa_i$ a specific realization. If $D$ is a causal graph, the joint distribution between the variables (vertices of the graph) factorizes relative to the graph as $p(x_1, x_2, \dots,) = \prod_i p(x_i|pa_i)$.

## 2 Rényi Common Entropy

We introduce the notion of Rényi common entropy, which generalizes the common entropy of [31].

**Definition 1.** *Rényi common entropy of order $r$ or Rényi$_r$ common entropy of two random variables $X, Y$ with probability distribution $p(x, y)$ is shown by $G_r(X, Y)$ and is defined as follows:*

$$
\begin{aligned}
G_r(X, Y) \coloneqq \min_{q(x,y,z)} \quad & H_r(Z) \\
s.t. \quad & I(X; Y|Z) = 0; \sum_z q(x, y, z) = p(x, y), \forall x, y; \sum q(.) = 1; q(.) \geq 0
\end{aligned} \tag{1}
$$

Rényi common entropy lower bounds the Rényi entropy of any variable that makes the observed variables conditionally independent. We focus on two special cases: Rényi$_0$ and Rényi$_1$ common entropies. Among all variables $Z$ such that $X \perp\!\!\!\perp Y|Z$, Rényi$_0$ common entropy is the logarithm of the minimum number of states of $Z$ and Rényi$_1$ common entropy is its minimum entropy.

In Section 3, we show that Rényi$_0$ common entropy can be used for distinguishing the latent graph from the triangle or direct graphs in Figure 1. Since we expect Rényi$_0$ common entropy to be sensitive to finite-sample noise in practice, we focus on Rényi$_1$ common entropy. Rényi$_1$ common entropy, or simply common entropy, was introduced in [31], where authors derived the analytical expression for two binary variables. They also remark that finding common entropy for non-binary variables is difficult. We propose an iterative update algorithm to approximate common entropy in practice, by assuming that we have access to the joint distribution between $X, Y$.

### LatentSearch: An Algorithm for Calculating Rényi$_1$ Common Entropy

In this section, our objective is to solve a relaxation of the Rényi$_1$ common entropy problem in (1). Instead of enforcing conditional independence as a hard constraint of the optimization problem, we introduce conditional mutual information as a regularizer to the objective function. This allows us to discover a trade-off between two factors, the entropy $H(Z)$ of the third variable and the residual dependency between $X, Y$ after conditioning on $Z$, measured by $I(X; Y|Z)$. We then have the loss

$$
\mathcal{L} = I(X; Y|Z) + \beta H(Z). \tag{2}
$$

<center>3</center>

---

**Algorithm 1** LatentSearch: Iterative Update Algorithm

---

1: **Input:** Supports of $x, y, z$, $\mathcal{X}, \mathcal{Y}, \mathcal{Z}$, respectively. $\beta \geq 0$ used in (2). Observed joint $p(x, y)$. Initialization $q_1(z|x, y)$. Number of iterations $N$.
2: **Output:** Joint distribution $q(x, y, z)$
3: **for** $i \in [N]$ **do**
4:    *Form the joint:*
    $q_i(x, y, z) \leftarrow q_i(z|x, y)p(x, y), \forall x, y, z.$
5:    *Calculate:*
$$q_i(z|x) \leftarrow \frac{\sum\limits_{y \in \mathcal{Y}} q_i(x, y, z)}{\sum\limits_{y \in \mathcal{Y}, z \in \mathcal{Z}} q_i(x, y, z)}, \qquad q_i(z|y) \leftarrow \frac{\sum\limits_{x \in \mathcal{X}} q_i(x, y, z)}{\sum\limits_{x \in \mathcal{X}, z \in \mathcal{Z}} q_i(x, y, z)}, \qquad q_i(z) \leftarrow \sum\limits_{x \in \mathcal{X}, y \in \mathcal{Y}} q_i(x, y, z)$$
6:    *Update:*
    $q_{i+1}(z|x, y) \leftarrow \frac{1}{F(x,y)} \frac{q_i(z|x)q_i(z|y)}{q_i(z)^{1-\beta}}$, where $F(x, y) = \sum\limits_{z \in \mathcal{Z}} \frac{q_i(z|x)q_i(z|y)}{q_i(z)^{1-\beta}}$.
7: return $q(x, y, z) \coloneqq q_{N+1}(z|x, y)p(x, y)$

---

Rather than searching over $q(x, y, z)$ and enforcing the constraint $\sum_z q(x, y, z) = p(x, y), \forall x, y$, we can search over $q(z|x, y)$ and set $q(x, y, z) = q(z|x, y)p(x, y)$. Therefore we have $\mathcal{L} = \mathcal{L}(q(z|x, y))$. The support size of $Z$ determines the number of optimization variables. Proposition 5 in [31] shows that without loss of generality, we can assume $|\mathcal{Z}| \leq |\mathcal{X}||\mathcal{Y}|$. In general, $\mathcal{L}$ is neither convex nor concave. Although first order methods (e.g., gradient descent) can be used to find a stationary point, as we empirically observe the convergence is slow and the performance is very sensitive to the step size. To this end, we propose a multiplicative update algorithm *LatentSearch* in Algorithm 1. Given $p(x, y)$, *LatentSearch* starts from a random initialization $q_0(z|x, y)$, and at each step $i$ iteratively updates $q_i(z|x, y)$ to $q_{i+1}(z|x, y)$ to minimize the loss (2). Specifically, in the $i^{th}$ step it marginalizes the joint $q_i(x, y, z)$ to get $q_i(z|x), q_i(z|y)$, and $q_i(z)$, and imposing a scaled product form on these marginals, updates the joint to return $q_{i+1}(x, y, z)$. This decomposition and the update rule are motivated by the partial derivatives associated with the Lagrangian of the loss function (2) (See Section 7.3). More formally, as we show in the following theorem, after convergence *LatentSearch* outputs a stationary point of the loss function. For the proof, please see Sections 7.3, 7.4.

**Theorem 1.** *The stationary points of LatentSearch are also stationary points of the loss in (2). Moreover, for $\beta = 1$, LatentSearch converges to either a local minimum or a saddle point of (2), unless it is initialized at a local maximum.*

Therefore, if the algorithm converges to a solution, it outputs either a local minimum, local maximum or a saddle point. We observe in our experiments that the algorithm always converges for $\beta \leq 1$.

For each $\beta$, *LatentSearch* outputs a distribution $q(.)$ from which $H(Z)$ can be calculated. When using *LatentSearch* to approximate Rényi$_1$ common entropy, we will run it for multiple $\beta$ values and pick the distribution $q(.)$ with the smallest $H(Z)$ such that $I(X; Y|Z) \leq \theta$ for a practical threshold $\theta$ to declare conditional independence. See Figure 3b for a sample output of *LatentSearch* for multiple $\beta$ values in the $I - H$ plane. See also lines $6 - 7$ of Algorithm 2 for an algorithmic description.

## 3 Identifying Correlation without Causation via Rényi Common Entropy

Suppose we observe two discrete random variables $X, Y$ to be statistically dependent. Reichenbach's common cause principle states that $X$ and $Y$ are either causally related, or there is a common cause that induces the correlation.[1] If the correlation is *only* due to a common cause, intervening on either variable will have no effect on the other. Therefore, it is important for policy decisions to identify this case of *correlation without causation*. Specifically, we want to distinguish latent graph from the triangle or direct graphs in Figure 1. Since our goal is not to identify the causal direction between $X$ and $Y$, we use triangle, direct and mediator graphs to refer to either direction. We show that, under certain assumptions, Rényi common entropy can be used to solve this identification problem.

Our key assumption is that the latent confounders, if they exist, have small Rényi entropy. In other words, in Figure 1 $H_r(Z) \leq \theta_r$ for some $\theta_r$. We consider two cases: Rényi$_0$ and Rényi$_1$ entropies. $H_0(Z) \leq \theta_0$ is equivalent to upper bounding the support size of $Z$. $H_1(Z) \leq \theta_1$ upper bounds the Shannon entropy of $Z$. In general, $H_1(Z) \leq \theta$ can be seen as a relaxation of $H_0(Z) \leq \theta$ as the latter

implies the former but not vice verse. Accordingly, we show stronger results for Rényi$_0$, whereas we leave the most general identifiability result of Rényi$_1$ as a conjecture. We also quantify how small the confounder's Rényi entropy should be for identifiability.

Note that bounding the Rényi entropy of the latent confounder in the latent graph bounds the Rényi common entropy of the observed variables. Therefore, in order to distinguish the latent graph from the triangle and direct graphs, we need to obtain lower bounds on the Rényi common entropy of a typical pair $X, Y$ when data is generated from the triangle or direct graphs.

We first establish bounds on the Rényi$_0$ common entropy for the triangle and direct graphs, which hold for almost all parametrizations. To measure the fraction of causal models for which our bound is valid, we use a natural probability measure on the set of joint distributions by sampling each conditional uniformly randomly from the probability simplex:

**Definition 2** (Uniform generative model (UGM)). *For any causal graph, consider the following generative model for the joint distribution $p(x_1, x_2, \ldots) = \prod_i p(x_i|pa_i)$, where $X_i \in \mathcal{X}_i, \forall i$: For all $i$ and $pa_i$, let the conditional distribution $p(X_i|pa_i)$ be sampled independently and uniformly randomly from the probability simplex in $|\mathcal{X}_i|$ dimensions.*

The following theorem uses the measure induced by UGM to show that for almost all distributions obtained from the triangle or direct graph, Rényi$_0$ common entropy of the observed variables is large.

**Theorem 2.** *Consider the random variables $X, Y, Z$ with supports $[m], [n], [k]$, respectively. Let $p(x, y, z)$ be a pmf sampled from the triangle or the direct graphs according to UGM. Then with probability 1, $G_0(X, Y) = \log(\min\{m, n\})$.*

Now consider the latent graph where $Z$ is the true confounder. We clearly have that $G_0(X, Y) \leq H_0(Z)$ since $Z$ indeed makes $X, Y$ conditionally independent. In other words, $G_0(X, Y)$ is upper bounded in the latent graph whereas it is lower bounded in the triangle and the direct graphs by Theorem 2. Therefore, as long as the correlation cannot be explained by the mediator graph, $G_0(X, Y)$ can be used as a parameter to identify the latent graph. In order to formalize this identifiability statement, we need two assumptions with parameters $(r, \theta)$:

**Assumption 1** $(r, \theta)$. *Consider any causal model with observed variables $X, Y$. Let $Z$ represent the variable that captures all latent confounders between $X, Y$. Then $H_r(Z) < \theta$.*

**Assumption 2** $(r, \theta)$. *Consider a causal model where $X$ causes $Y$. If $X$ causes $Y$ only through a latent mediator $Z$, i.e., $X \rightarrow Z \rightarrow Y$, then $H_r(Z) \geq \theta$.*

Assumption 1 states that the collection of latent confounders, represented by $Z$, has to be "simple", which is quantified by its Rényi entropy. This assumption can also be interpreted as relaxing the causal sufficiency assumption by allowing *weak* confounding. Assumption 2 states that if the correlation is induced only due to a mediator, this mediator cannot have low Rényi entropy. Even though this assumption might seem restrictive, we provide evidence on both real and synthetic data in Section 5.2 to show it indeed often holds in practice. We have the following corollary:

**Corollary 1.** *Consider the random variables $X, Y$ with supports $[m], [n]$, respectively. For $\theta = \log(\min\{m, n\})$, latent graph can be identified with probability 1 under UGM, Assumption 1,2$(0, \theta)$.*

Corollary 1 indicates that, when the latent confounder has less than $\min\{m, n\}$ number of states, we can infer that the true causal graph is the latent graph from observational data under Assumptions 1 and 2. However, using common entropy, we cannot distinguish triangle graph from the direct graph. Also note that the identifiability result holds for *almost all* parametrizations of these graphs, i.e., the set of parameters where it does not hold has Lebesgue measure zero.

Next, we investigate if Rényi$_1$ common entropy can be used for the same goal. Finding Rényi$_1$ common entropy in general is challenging. For binary $X, Y$ we can use the analytical expression of [31] to show that $G_1(X, Y)$ is almost always larger than $H(Z)$ asymptotically for the triangle graph:

**Theorem 3.** *Consider the random variables $X, Y, Z$ with supports $[2], [2], [k]$, respectively. Let $p(x, y, z)$ be a pmf sampled from the triangle graph according to UGM except $p(z)$, which can be arbitrary. Then $\lim_{H(Z) \rightarrow 0} \mathbb{P}(G_1(X, Y) > H(Z)) = 1$, where $\mathbb{P}$ is the probability measure induced by UGM.*

In Section 7.8, we provide simulations for binary and ternary $Z$ to demonstrate the behavior for small non-zero $H(Z)$. Then, we have the following asymptotic identifiability result using common entropy:

---

**Algorithm 2** InferGraph: Identifying the Latent Graph

---

1: **Input:** $k$ : Support size of $Z$, $p(x,y)$, $T : I(X;Y|Z)$ threshold, $\{\beta_i\}_{i \in [N]}$, $\theta : H(Z)$ threshold.
2: Randomly initialize $N$ distributions $q_0^{(i)}(z|x,y), i \in [N]$.
3: **for** $i \in [N]$ **do**
4:     $q^{(i)}(x,y,z) \leftarrow LatentSearch(q_0^{(i)}(z|x,y), \beta_i)$.
5:     Calculate $I^{(i)}(X;Y|Z)$ and $H^{(i)}(Z)$ from $q^{(i)}(x,y,z)$.
6: $S = \{i : I^{(i)}(X;Y|Z) \leq T\}$.
7: **if** $\min(\{H^{(i)}(Z) : i \in S\}) > \theta$ or $S = \emptyset$ **then**
8:     **return** Triangle or Direct Graph
9: **else**
10:     **return** Latent Graph

---

---

**Algorithm 3** EntropicPC (for $F = False$) and EntropicPC-C (for $F = True$)

---

1: **Input:** CI Oracle $\mathcal{C}$ for $\mathcal{V} = \{X_1, \ldots X_n\}$. Common entropy oracle $\mathcal{B}$. Entropy oracle $H$. Flag $F$.
2: Form the complete undirected graph $D = (\mathcal{V}, \mathcal{E})$ on node set $\mathcal{V}$.
3: $l \leftarrow -1$. $maysep(X_i, X_j) \leftarrow True, \forall i, j$.
4: **while** $\exists i, j$ s.t. $(X_i, X_j) \in \mathcal{E}$ **and** $|adj_D(X_i) \backslash \{X_j\}| > l$ **and** $maysep(X_i, X_j) = True$. **do**
5:     $l \leftarrow l + 1$
6:     **for** All $i, j$ s.t. $(X_i, X_j) \in \mathcal{E}$ and $|adj_D(X_i) \backslash \{X_j\}| \geq l$ **do**
7:         **while** $(X_i, X_j) \in \mathcal{E}$ and $\exists S \subseteq adj_D(X_i) \backslash \{X_j\}$ s.t. $|S| = l$ **and** $maysep(X_i, X_j) = True$. **do**
8:             Pick a new $S \subseteq adj_D(X_i) \backslash \{X_j\}$ s.t. $|S| = l$.
9:             **if** $\mathcal{C}(X, Y|Z) = True$ **then**
10:                 **if** $H(S) \geq \mathcal{B}(X_i, X_j)$ **then**
11:                     $\mathcal{E} \leftarrow \mathcal{E} - \{(X_i, X_j)\}$.
12:                     $sepset(X_i, X_j) \leftarrow S$.
13:                 **else if** $F = True$ **then**
14:                     $maysep(X_i, X_j) \leftarrow False$
15:                 **else**
16:                     **if** $\mathcal{B}(X_i, X_j) \geq 0.8 \min\{H(X_i), H(X_j)\}$ **then**
17:                         $maysep(X_i, X_j) \leftarrow False$
18: Orient unshielded colliders according to separating sets $\{sepset(X_i, X_j)\}_{i,j \in [n]}$ [11].
19: Orient as many of the remaining undirected edges as possible by repeatedly applying the Meek rules [35].
20: **Return:** $D = (\mathcal{V}, \mathcal{E})$.

---

**Corollary 2.** *For binary $X, Y$ under UGM, Assumption 1,2($1, \theta$) if the entropy upper bound $\theta$ is known, the fraction of causal models for which latent graph can identified goes to $1$ as $\theta$ goes to $0$.*

For the general case, we conjecture that when the data is sampled from the triangle or the direct graphs, $G_1(X, Y)$ scales with $\min\{H(X), H(Y)\}$.

**Conjecture 1.** *Consider the random variables $X, Y, Z$ with supports $[m], [n], [k]$, respectively. Let $p(x, y, z)$ be a pmf sampled from the triangle or direct graphs according to UGM except $p(z)$, which can be arbitrary. Then, there exists a constant $\alpha = \Theta(1)$ such that with probability $1 - (\min\{m, n\})^{-c}$ $G_1(X, Y) > \alpha \min\{H(X), H(Y)\}$ for some constant $c = c(\alpha)$.*

According to Conjecture 1, we expect that for most of the parametrizations of the triangle and direct graphs, common entropy of the observed variables should be lower-bounded by the entropies of the observed variables, up to a scaling by a constant. It is easy to see that under assumptions similar to those in Corollaries 1 and 2, Conjecture 1 implies identifiability of the latent graph. In Section 5, we conduct experiments to support the conjecture and identify $\alpha$. We conclude this section by formalizing how *LatentSearch* can be used in Algorithm 2, under Assumption $1(1, \theta)$, $2(1, \theta)$. Conjecture 1 suggests that, in Algorithm 2, we can set $\theta = \alpha \min\{H(X), H(Y)\}$ for some $\alpha < 1$.

## 4 Entropic Constraint-Based Causal Discovery

A causal graph imposes certain CI relations in the data. Constraint-based causal discovery methods utilize CI statements to reverse-engineer the underlying causal graph. Consider a causal graph over a set $\mathcal{V}$ of observed variables. Constraint-based methods identify a set $S_{X,Y} \subset \mathcal{V}$ as a *separating set*

for every pair $X, Y$ if the CI statement $X \perp\!\!\!\perp Y \mid S_{X,Y}$ holds in the data. Starting with a complete graph, edges between pairs are removed if they are separable by some set. Separating sets are later used to orient parts of the graph, which is followed by a set of orientation rules [47, 55] .

Despite being grounded theoretically in the large sample limit, in practice these methods require a large number of samples and are very sensitive to noise: An incorrect CI statement early on might lead to a drastically different causal graph at the end due to their sequential nature. Another issue is that the distribution should be faithful to the graph, i.e., any connected pair should be dependent [47, 29].

To help alleviate some of these issues, we propose a simple modification to the existing constraint-based learning algorithms using common entropy. Our key observation is that the common entropy of two variables provide an information-theoretic lower bound on the entropy of *any* separating set. In other words, common entropy provides us with a necessary condition for a set $S_{X,Y}$ to be a valid separating set: $X \perp\!\!\!\perp Y \mid S_{X,Y}$ only if $H(S_{X,Y}) \geq G_1(X, Y)$. Accordingly, we can modify any constraint-based method to ensure this condition. We only lay out our modifications on the PC algorithm. It can be trivially applied to other methods such as modern variants of PC and FCI.

We propose two versions: *EntropicPC* and the conservative version *EntropicPC-C*. In both, $S_{X,Y}$ is accepted only if $H(S_{X,Y}) \geq G_1(X, Y)$. The difference is how they handle pairs $X, Y$ that are deemed CI despite that $H(S_{X,Y}) < G_1(X, Y)$. EntropicPC-C concludes that the data for $X, Y$ is unreliable and simply declares them non-separable by any set. EntropicPC only does this when common entropy is large, i.e., $G_1(X, Y) \geq 0.8 \min\{H(X), H(Y)\}$; otherwise it searches for another set that may satisfy $H(S_{X,Y}) \geq G_1(X, Y)$. 0.8 is chosen based on our experiments in Section 5.

We provide the pseudo-code in Algorithm 3. It is easy to see that both algorithms are sound in the sample limit. The case of $S = \emptyset$ in **line 10** is of special interest. Setting $H(\emptyset) = 0$ is not reasonable with finitely many samples since the common entropy of independent variables will not be estimated as exactly zero. To address this, in simulations $H(\emptyset)$ is set to $0.1 \min\{H(X), H(Y)\}$ in line 10. This and the choice of $0.8$ as the coefficient in **line 16** can be seen as hyper-parameters to be tuned.

## 5 Experiments

### 5.1 Performance of LatentSearch

We evaluate how well *LatentSearch* performs by generating data from the latent graph and comparing the entropy it recovers with the entropy of the true confounder $Z$ in Figure 2a. In the generated data, we ensure $H(Z)$ is bounded above by 1 for all $n$. This makes the task harder for the algorithm for larger $n$. The left axis shows the fraction of times *LatentSearch* recovers a latent with entropy smaller than $H(Z)$. The right axis shows the worst-case performance in terms of the entropy gap between the algorithm output and true entropy. We generated $100$ random distributions for each $n$. The same number of iterations is used for the algorithm for all $n$. As expected, performance slowly degrades as $n$ is increased, since $H(Z) < 1, \forall n$. We conclude that *LatentSearch* performs well within the range of $n$ values we use in this paper. Further research is needed to make *LatentSearch* adapt to $n$.

### 5.2 Validating Assumption 2: No Low-Entropy Mediator

We conducted synthetic and real experiments to validate the assumption that, in practice, it is unlikely for cause-effect pairs to only have low-entropy mediators. First, in Figure 3c we generated data from $X \to Z \to Y$ and evaluated $H(Z)$. $p(X)$ is sampled uniformly from the probability simplex. $p(Z|x), \forall x$ are sampled from Dirichlet with parameter $\alpha_{Dir}$. It is observed that mediator entropy scales with $\log_2(n)$ for all cases. This supports Assumption 2 by asserting that for most causal models, unless the mediator has a constant number of states, its entropy is close to $H(X), H(Y)$.

Second, in Figure 3a, we run *LatentSearch* on the real cause-effect pairs from Tuebingen dataset [36]. Our goal is to test if the causation can be solely due to low-entropy mediators: If it is, then common entropy should be small since mediator can make the observed variables conditionally independent. We used different thresholds for conditional mutual information for declaring two variables conditionally independent. Investigating typical $I - H$ plots for this dataset (see (b)), we conclude that $0.001$ is a suitable CMI threshold for this dataset. From the empirical cdf of $\alpha := \frac{G_1(X,Y)}{\min\{H(X),H(Y)\}}$ across the dataset, we identified that for most pairs $G_1(X, Y) \geq 0.8 \min\{H(X), H(Y)\}$. This indicates that if the causation is solely due to a mediator, it must have entropy of at least $0.8 \min\{H(X), H(Y)\}$.

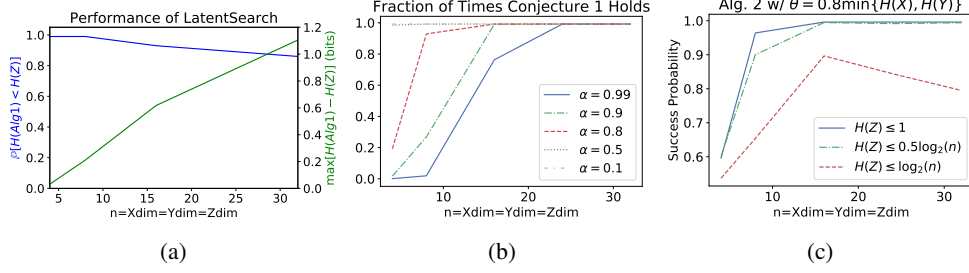

(a)                          (b)                          (c)

Figure 2: **(a)** [Section 5.1] Performance of *LatentSearch* (Alg. 1) on synthetic data. **(b)** [Section 5.3] Fraction of times Conjecture 1 holds in synthetic data for different values of $\alpha$. **(c)** [Section 5.4] Performance of Algorithm 2 on synthetic data for different confounder entropies.

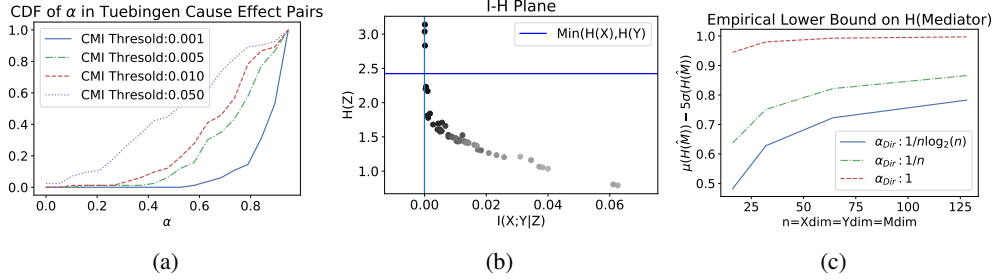

(a)                          (b)                          (c)

Figure 3: **(a)** [Section 5.3,5.2] Empirical cumulative distribution function for $\alpha$ from Conjecture 1 from Tuebingen Data for various conditional mutual information thresholds. **(b)** Tradeoff curve discovered by *LatentSearch* for pair 7 of Tuebingen. Each point is the output of *LatentSearch* for a different value of $\beta$. The sharp elbow around $I(X;Y|Z) = 0$ hints that the suitable conditional mutual information threshold for this pair is $\approx 0.001$. **(c)** Entropy of mediator in synthetic data. $y-$axis shows the mean minus $5$ times the st-dev. of entropy as a fraction of $\log_2(n)$ as an empirical lower bound. Results show that, for almost all models, entropy of the mediator scales with $\log_2(n)$.

## 5.3 Validating Conjecture 1

In Figure 2b, for each $n$ we sample 1000 distributions from the triangle graph with $H(Z) \leq 1$. Even with a 1 bit latent, we observe that $G_1(X,Y)$ scales with $\min\{H(X), H(Y)\}$. In fact the result hints at the even stronger statement that that for any $\alpha$, $\exists n_0(\alpha)$ where the conjecture holds $\forall n \geq n_0(\alpha)$.

Experiments with the Tuebingen data, as explained in Section 5.2 and Figure 3a, similarly supports Conjecture 1: If a pair is causally related, common entropy typically scales with $\min\{H(X), H(Y)\}$.

## 5.4 Identifiability via Algorithm 2

In Figure 2c we uniformly mixed data from triangle and latent graphs with different entropy bounds on the confounder. Since in a practical problem, true $H(Z)$ will not be available, we used $0.8 \min\{H(X), H(Y)\}$ as the entropy threshold $\theta$ in Algorithm 2. The results indicate that even if the true $H(Z)$ scales with $n$ (e.g., $0.5 \log_2(n)$), we can still distinguish latent from triangle graph. $\leq$ can be interpreted as $\approx$ due to our sampling method of $p(z)$, as detailed in Section 7.10.

## 5.5 Evaluating EntropicPC

In this section, to illustrate the performance improvement provided by using common entropy, we compare the order-independent version of PC algorithm [11] with our proposed modification EntropicPC and its conservative version EntropicPC-C as described in Algorithm 3 on both synthetic graphs and data generated from them and on ADULT dataset. We use *pcalg* package in R [24, 18]

**Synthetic Graphs/Data:** In Figure 4 we randomly sampled joint distributions from 100 Erdös-Rényi graphs on 10 nodes (see Section 7.12 for 3-node graphs) each with 8 states, and obtained datasets with $1k, 5k, 80k$ samples. $80k$ is chosen to mimic the infinite-sample regime where we expect identical

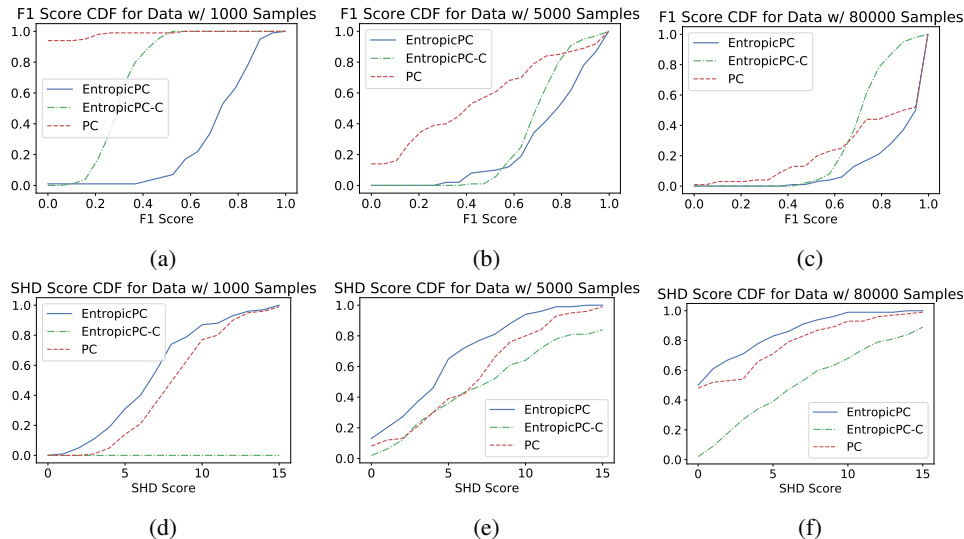

Figure 4: [Section 5.5] Performance of EntropicPC, EntropicPC-C and PC in synthetic data.

performance with PC. **(a), (b), (c)** shows the empirical cumulative distribution function (CDF) of F1 score for identifying edges in either direction correctly (skeleton). Since $F_1 = 1$ is ideal, the best-case CDF is a step function at $1$. Accordingly, lower the CDF curve the better. **(d), (e), (f)** shows the empirical CDF of Structural Hamming Distance (SHD) [49] between the true essential graphs, and the output of each algorithm using the same synthetic data. Since $SHD = 0$ is ideal, the best-case CDF is a step function at $0$, Accordingly, higher the CDF curve the better. EntropicPC provides very significant improvements to skeleton discovery as indicated by the F1 score and moderate improvement to identifying the true essential graph as indicated by SHD in the small-sample regime.

**On ADULT Dataset:** We compare the performance of EntropicPC with the baseline PC algorithm that we used for our modifications. Entropic PC identifies the following additional edges that are missed by the PC algorithm: It discovers that *Marital Status, Relationship, Education, Occupation* causes *Salary*. Even though there is no ground truth, at the very least, we expect *Education* and *Occupation* to be causes of *Salary*, whereas PC outputs *Salary* as an isolated node, implying it is not caused by any of the variables. We believe the reason it that for every neighbor, there is some conditioning set and a configuration with very few number of samples. This can easily be interpreted as conditional independence by a CI tester. EntropicPC alleviates this problem by concluding that there is significant dependence between these variables by checking their common entropy. From both real and synthetic simulations, we conclude that common entropy is more robust to small number of samples than CI testing. For space restriction, causal graphs are shown in Figure 8 in Appendix.

## 6 Conclusions

In this paper, we showed that common entropy, an information-theoretic quantity, can be used for causal discovery. We introduced the notion of Rényi common entropy and proposed a practical algorithm for approximately calculating $\text{Rényi}_1$ common entropy. Next, we showed theoretically that common entropy can be used to identify if the observed variables are causally related under certain assumptions. Finally, we showed that common entropy can be used to improve the existing constraint-based causal discovery algorithms. We evaluated our results on synthetic and real data.

## Acknowledgments and Disclosure of Funding

This work was supported in part by NSF Grants SATC 1704778, CCF 1763702, 1934932, AF 1901292, 2008710, 2019844, 1731754, 1646522, 1609279, ONR, ARO W911NF-17-1-0359, research gifts by Western Digital, WNCG IAP, computing resources from TACC, the Archie Straiton Fellowship.

## Broader Impact

This work lays the foundations for using the information-theoretic notion of common entropy within the context of discovering causal relations from data.

Problems that require discovering causal relations from observational data are prominent across many different fields. Causality is also central to the development of AI. Therefore, we expect this work to have a positive impact by providing a new methodology and identifying settings in which causality can be inferred using this framework.

In terms of the negative effects, we do not foresee an immediate negative effect that may arise because of this work. The only risks would be due to the risks associated with having stronger machine learning models, and better AI that could be misused or exploited.

## Footnotes

[1]We assume no selection bias in this work, which can also induce spurious correlation.

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
