[Supplementary Material]

# 7 Appendix

## 7.1 Detailed Background

Let $D = (\mathcal{V}, \mathcal{E})$ be a directed acyclic graph on the set of vertices $\mathcal{V} = \{V_1, V_2, \ldots, V_n\}$ with directed edge set $\mathcal{E}$. Each directed edge $e_k \in \mathcal{E}$ is a tuple $e_k = (V_i, V_j)$. Let $\mathbb{P}$ be a joint distribution over a set of variables labeled by $\mathcal{V}$. $D$ is called a valid Bayesian network for the distribution $\mathbb{P}$, if $\mathbb{P}$ factorizes with respect to the graph $D$ as $\mathbb{P}(V_1, V_2, \ldots V_n) = \prod_i \mathbb{P}(V_i | pa_i)$, where $pa_i$ are the set of parents of vertex $V_i$ in graph $D$. In a Bayesian network $D$ that is valid for $\mathbb{P}$, if three vertices $X, Y, Z$ satisfy a graphical criterion called the *d-separation* on $D$, then $X \perp\!\!\!\perp Y \,|\, Z$ in $\mathbb{P}$. The faithfulness assumption allows us to infer dependence relations based on d-separation: $\mathbb{P}$ is said to be faithful to graph $D$ when the following holds: Any three variables $X, Y, Z$ that are not d-separated are conditionally dependent, i.e., $X \not\!\perp\!\!\!\perp Y \,|\, Z$ in $\mathbb{P}$.

Note that the edges in a Bayesian network do not carry a physical meaning: They simply indicate how a joint distribution can be factorized. *Causal Bayesian networks* (or causal graphs) [38] on the other hand capture the causal relations between variables: They extend the notion of Bayesian networks to different experimental, the so called interventional settings. *An intervention* is an experiment that changes the workings of the underlying system and sets the value of a variable, shown as $do(X = x)$. Causal Bayesian networks allow us to calculate the joint distributions under these experimantal conditions, called the interventional distributions[2].

In this paper, we work with the causal graphs given in Figure 1. From the d-separation principle, we see that the latent graph satisfies $X \perp\!\!\!\perp Y \,|\, Z$, whereas under the faithfulness condition, $X \not\!\perp\!\!\!\perp Y \,|\, Z$ in the triangle graph or the direct graph. Checking the existence of such a latent variable can help us recover the true causal graph as we discover in the next sections. We work with discrete ordinal or categorical variables. Suppose the support sizes of the observed variables $X$ and $Y$ are $m$ and $n$, respectively. The joint distribution can be represented with an $m \times n$ non-negative matrix whose entries sum to 1. We assume that we have access to this joint distribution.

We use $[n]$ to represent the set $\{1, 2, \ldots, n\}$ for any $n \in \mathbb{N}$. Capital letters represent random variables, lowercase letters represent realizations[3]. Letters $X, Y$ are reserved for the observed variables, whereas $Z$ is used for the latent variable. To represent the probability mass function over three variables $X, Y, Z$, we use $p(x, y, z) := \mathbb{P}(X = x, Y = y, Z = z)$ and similarly for any conditional $p(z|x, y) := \mathbb{P}(Z = z | X = x, Y = y)$. For a function $q(x, y, z)$ that is understood to be a probability mass function, we use shorthand notation for marginals and conditionals such as $q(x, y)$ and $q(x|z)$ to represent the functions obtained from $q(x, y, z)$ via standard operations on probability distributions. Lowercase boldface letters are used for vectors and uppercase boldface letters are used for matrices. We also use $p(Z|x, y)$ to represent the conditional distribution $\mathbb{P}(Z|X = x, Y = y)$ (Similarly for $p(Z|x), p(Z|y)$). $\mathrm{card}(X)$ stands for the support size of $X$. Rényi entropy of order $\alpha$ of a random variable $X$ is defined as $H_\alpha(X) = \frac{1}{1-\alpha} \log \sum_i p_i^\alpha$. Rényi entropy of order 0 gives the support size of a random variable. It can be shown that in the limit as $\alpha \to 1$, Rényi entropy becomes Shannon entropy, defined as $H_1(X) = -\sum_x p(x) \log_2(p(x))$ in bits. In a graph $D$ with nodes labeled as $\{X_i\}_i$, $pa_i$ stands for the set of parents of $X_i$ in $D$. $Dir(\alpha)$ stands for Dirichlet distribution with parameter $\alpha$.

## 7.2 Detailed Related Work

**Latent Variable Discovery:** Latent variables have been used to model and explain dependence between observed variables in different communities under different names. Probabilistic latent semantic analysis (pLSA) [19] aims at constructing a variable that explains dependence. However the objective is not to minimize entropy of the constructed variable. Latent Dirichlet allocation (LDA) is another framework which is widely used in topic modeling [4, 2]. Although LDA encourages sparsity of topics, this does not correspond to minimizing the support size of the constructed latent variable. Factorizing the joint distribution matrix between two observed variables via NMF with generalized KL divergence loss recovers solutions to the pLSA problem [16]. Similar to pLSA, NMF does not have an incentive to discover low-entropy latents.

Perhaps the most relevant to ours in the machine learning literature are the two papers in the Bayesian setting [5, 45]. They use low-entropy priors on the latent variable's distribution while performing inference. However their approach is different and their methods cannot be used to discover the tradeoff between conditional mutual information and the entropy of the latent variable. In [45], the authors use low-entropy prior as a proxy for discovering latent factors with sparse support.

Finding the latent variable with smallest entropy that renders the two observed variables conditonally independent is closely related to some of the problems in information theory: Wyner's common information [53] is defined as the minimum rate of the source from which the observed variables $X, Y$ can be reconstructed using additional random bits. Wyner allows multiple channel uses and is interested in the approximate reconstruction of the observed joint distribution. This can be seen as approximate reconstruction of the joint distribution when we raise the dimension of $X$ and $Y$ via cartesian product with itself. [31] considers finding the source with the minimum rate for the exact recovery of the observed joint distribution, but still in the asymptotic regime of multiple channel uses. They also introduce the notion of common entropy and obtain an analytical expression for binary variables, which we utilize in this work.

**Learning Causal Graphs with Latents:** Learning causal graphs with latent variables has been extensively studied in the literature. In graphs with many observed variables, some of the edges can be recovered from the observational data (for example through algorithms that employ conditional independence (CI) tests such as IC* [38] and FCI [47]). However, latent variables make the CI tests less informative, by inducing spurious correlations between the observed variables. For example for the graphs in Figure 1 CI tests on the observed variables is not informative of the causal structure.

Identifiability of causal structures without latent variables from data has been studied extensively in the literature under various assumptions [20, 41, 37, 39, 40, 3, 15, 26]. Our approach can be seen as an extension of [26]: There, the authors assume that the exogenous variables have small Rényi entropy and suggest an algorithm to distinguish the causal graph $X \to Y$ from $X \leftarrow Y$. However, their approach cannot be used in the presence of latent variables. In the presence of latents [6] considers a setup similar to [26], where the hidden variable has small support size, however also assumes the mapping to the hidden variable is deterministic. In [23], authors identify a condition on $p(Y|X)$ which implies that there does not exist any latent variable $Z$ with small support which can make $X, Y$ conditionally independent. For discrete variables, this assumption implies that the conditionals $p(Y|x)$ lie on the boundary of the probability simplex, which corresponds to the joint probability matrix to be sparse in a structured way. In the continuous variable setting, [43] propose using kernel methods to detect latent confounders. [52] and [8] analyzes the discoverability of causal structures with latents using the entropic vector of the variables. Finally, related work also includes [21] and [33], where the authors extend the additive noise model based approach in [20] to the case with a latent confounder. Algebraic geometry can be used to distinguish causal graphs as the set of distributions that can be encoded by a graph correspond to different algebraic varieties. However, these methods in general are not scalable beyond a few variables and a few number of states [32]. Authors in [22] propose using Kolmogorov complexity of the causal model and declare the graph with smaller complexity to be the true graph. [25] uses description length as a proxy to Kolmogorov complexity to identify the latent confounders. In [48], the authors use information inequalities to infer which subsets of a set of observed variables must have latent confounders, along with an associated lower bound on the entropy of these confounders. In our setting of two observed variables, this gives the trivial bound of $H(Z) \geq I(X;Y)$ for any latent confounder $Z$.

### 7.3 Proof of Theorem 1: Stationarity

In this section, we show the first part of Theorem 1, i.e., that the stationarity points of the algorithm are also stationary points of the given loss function. We write the objective function more explicitly in terms of the optimization variables $q(z|x, y)$:

$$\mathcal{L}(q(\cdot|\cdot, \cdot)) = \sum_{x,y,z} q(x, y, z) \log \left( \frac{q(x, y|z)}{q(x|z)q(y|z)} \right) - \beta \sum_z q(z) \log(q(z)) \tag{3}$$

$$= \sum_{x,y,z} p(x, y)q(z|x, y) \log \left( \frac{q(z|x, y)}{q(z|x)q(z|y)} \right) + (1 - \beta) \sum_z q(z) \log(q(z)) + I(X;Y) \tag{4}$$

by Bayes rule and assuming that $q(z|x, y)$ and $p(x, y)$ are strictly positive.

Our objective then is

$$\underset{q(z|x,y)}{\text{minimize}} \quad \mathcal{L}(q(z|x,y))$$

$$\text{subject to} \quad \sum_z q(z|x,y) = 1, \ \forall x, y, \tag{5}$$

$$q(z|x,y) \geq 0, \ \forall z, x, y.$$

We can write the Lagrangian, which we represent with $\bar{\mathcal{L}}$, as

$$\bar{\mathcal{L}} = \sum_{x,y,z} p(x,y)q(z|x,y) \log \left( \frac{q(z|x,y)}{q(z|x)q(z|y)} \right) + I(X;Y) + (1-\beta) \sum_z q(z) \log(q(z))$$

$$+ \sum_{x,y} \delta_{x,y} \left( \sum_z q(z|x,y) - 1 \right) \tag{6}$$

In order to find the stationary points of the loss, we take its first derivative and set it to zero. To compute the partial derivatives, notice that $q(z|x), q(z|y), q(z)$ are linear functions of $q(z|x,y)$ (use Bayes rule and marginalization). We can then easily write the partial derivatives of these quantities with respect to $q(z|x,y)$ as follows:

$$\frac{\partial q(z|x)}{\partial q(z|x,y)} = \frac{\partial \sum\limits_{y'} q(z|x,y')p(y'|x)}{\partial q(z|x,y)} = p(y|x),$$

$$\frac{\partial q(z|y)}{\partial q(z|x,y)} = \frac{\partial \sum\limits_{x'} q(z|x',y)p(x'|y)}{\partial q(z|x,y)} = p(x|y)$$

$$\frac{\partial q(z)}{\partial q(z|x,y)} = \frac{\partial \sum\limits_{x',y'} q(z|x',y')p(x',y')}{\partial q(z|x,y)} = p(x,y).$$

Using these expressions we have the following.

$$\frac{\partial \bar{\mathcal{L}}}{\partial q(z|x,y)} = p(x,y) \left[ 1 + \log(q(z|x,y)) - (1 + \log(q(z|x))) \right.$$

$$- (1 + \log(q(z|y))) \quad + (1-\beta)(1 + \log(q(z))) + \delta_{x,y} \right]$$

$$= p(x,y) \left[ -\beta + \delta_{x,y} + \log \left( \frac{q(z|x,y)q(z)^{1-\beta}}{q(z|x)q(z|y)} \right) \right]$$

Assuming $p(x,y) > 0$, any stationary point then satisfies

$$q(z|x,y) = \left( \frac{1}{2} \right)^{\delta_{x,y} - \beta} \frac{q(z|x)q(z|y)}{q(z)^{1-\beta}} \tag{7}$$

Since $q(z|x,y)$ is a probability distribution, we have

$$\sum_z q(z|x,y) = \left( \frac{1}{2} \right)^{\delta_{x,y} - \beta} \sum_z \frac{q(z|x)q(z|y)}{q(z)^{1-\beta}} = 1 \tag{8}$$

Defining $F(x,y) := \left( \frac{1}{2} \right)^{\delta_{x,y} - \beta}$, we have

$$F(x,y) = \frac{1}{\sum\limits_z \frac{q(z|x)q(z|y)}{q(z)^{1-\beta}}}. \tag{9}$$

From the algorithm description, any stationary point of Algorithm 1 should satisfy

$$q(z|x,y) = F(x,y) \frac{q(z|x)q(z|y)}{q(z)^{1-\beta}}, \tag{10}$$

for the same $F(x,y)$ defined above. Therefore a point is a stationary point of the loss function if and only if it is a stationary point of LatentSearch (Algorithm 1). $\qquad \square$

## 7.4  Proof of Theorem 1: Convergence

In this section, we show the latter statement in Theorem 1, i.e., *LatentSearch* converges to a local minimum or a saddle point. We can rewrite the loss as

$$\mathcal{L}(q(\cdot|\cdot,\cdot)) = \sum_{x,y,z} q(x,y,z) \log\left(\frac{q(x,y|z)}{q(x|z)q(y|z)}\right) - \beta \sum_z q(z) \log(q(z)) \tag{11}$$

$$= \sum_{x,y,z} p(x,y)q(z|x,y) \log\left(\frac{q(z|x,y)}{q(z|x)q(z|y)}\right)$$

$$+ I(X;Y) + (1-\beta) \sum_z q(z) \log(q(z)), \tag{12}$$

If we substitute $\beta = 1$, we obtain

$$\mathcal{L}(q(\cdot|\cdot,\cdot)) = \sum_{x,y,z} p(x,y)q(z|x,y) \log\left(\frac{q(z|x,y)}{q(z|x)q(z|y)}\right) + I(X;Y). \tag{13}$$

Our optimization problem can be written as

$$\begin{aligned} \underset{q(z|x,y)}{\text{minimize}} \quad & \mathcal{L}(q(z|x,y)) \\ \text{subject to} \quad & \sum_z q(z|x,y) = 1, \forall x, y. \end{aligned} \tag{14}$$

Notice that $\mathcal{L}(q(z|x,y))$ is not convex or concave in $q(z|x,y)$. However we can rewrite the minimization as follows:

$$\begin{aligned} \underset{q(z|x,y)}{\text{minimize}} \quad \underset{r(z|x),s(z|y)}{\text{minimize}} \quad & \sum_{x,y,z} p(x,y)q(z|x,y) \\ & \log\left(\frac{q(z|x,y)}{r(z|x)s(z|y)}\right) + I(X;Y) \\ \text{subject to} \quad & \sum_z q(z|x,y) = 1, \forall x, y \\ & \sum_z r(z|x) = 1, \forall x, \\ & \sum_z s(z|y) = 1, \forall y. \end{aligned}$$

To see that (15) is equivalent to (14), notice that the optimum for the inner minimization is $r^*(z|x) = q(z|x)$ and $s^*(z|x) = q(z|y)$. This is due to the fact that (15) is convex in $r(z|x)$ and $s(z|y)$ and concave in $t(z)$, which can be seen through the partial derivatives of the Lagrangian:

$$\underset{q(z|x,y)}{\min} \quad \underset{r(z|x),s(z|y)}{\min} \quad \sum_{x,y,z} p(x,y)q(z|x,y) + \log\left(\frac{q(z|x,y)}{r(z|x)s(z|y)}\right) + I(X;Y) \tag{15}$$

$$+ \sum_{x,y} \delta_{x,y}\left(\sum_z q(z|x,y) - 1\right) + \sum_x \eta_x \left(\sum_z r(z|x) - 1\right) \tag{16}$$

$$+ \sum_x \nu_y \left(\sum_z s(z|y) - 1\right) \tag{17}$$

Let $\bar{\mathcal{L}}$ be defined as

$$\bar{\mathcal{L}} = \sum_{x,y} \delta_{x,y}\left(\sum_z q(z|x,y) - 1\right) + \sum_x \eta_x \left(\sum_z r(z|x) - 1\right) + \sum_x \nu_y \left(\sum_z s(z|y) - 1\right) \tag{18}$$

For fixed $q(z|x, y), s(z|y)$, we have

$$\frac{\partial \bar{\mathcal{L}}}{\partial r(z|x)} = -\frac{p(x)q(z|x)}{r(z|x)} + \eta_x$$

$$\frac{\partial^2 \bar{\mathcal{L}}}{\partial r(z|x)^2} = \frac{p(x)q(z|x)}{r(z|x)^2}.$$

Therefore $\bar{\mathcal{L}}$ is convex in $r(z|x)$ and the optimum can be obtained by setting the first derivative to zero. Then we have

$$r^*(z|x) = \frac{p(x)q(z|x)}{\eta_x}, \forall x, z. \tag{19}$$

Since we have $\sum_z r^*(z|x) = \frac{p(x)}{\eta_x} \sum_z q(z|x) = 1$, we obtain $r^*(z|x) = q(z|x)$. Similarly, we can show that $s^*(z|x) = q(z|y)$. Notice that this inner minimization is exactly the same as the first update of Algorithm 1.

We can also show that $\mathcal{L}$ is convex in the variables $r, s$ jointly: This can be seen through the fact that $\frac{\partial^2}{\partial r(z|x)s(z|y)}\mathcal{L} = 0$ and the Hessian is positive definite.

This concludes that (15) is equivalent to (5). Moreover, since the objective function is convex in $q(z|x, y)$ and also jointly convex in $r(z|x), s(z|y)$, we can switch the order of the minimization terms. Therefore, we can equivalently write

$$\min_{r(z|x),s(z|y)} \quad \min_{q(z|x,y)} \quad \sum_{x,y,z} p(x, y)q(z|x, y) \log\left(\frac{q(z|x, y)}{r(z|x)s(z|y)}\right) + I(X; Y) + \bar{\mathcal{L}} \tag{20}$$

Let us analyze the inner minimization in this equivalent formulation for fixed $r(z|x), s(z|x)$. Similarly, we can take the partial derivative as follows:

$$\frac{\partial \bar{\mathcal{L}}}{\partial q(z|x, y)} = p(x, y)\left[1 + \log(q(z|x, y)) - \log(r(z|x)) - \log(s(z|x)) + \delta_{x,y}\right]$$

$$= p(x, y)\left[1 + \delta_{x,y} + \log\left(\frac{q(z|x, y)}{r(z|x)s(z|y)}\right)\right]$$

$$\frac{\partial^2 \bar{\mathcal{L}}}{\partial q(z|x, y)^2} = p(x, y)\left[\frac{1}{q(z|x, y)}\right].$$

Notice that $\frac{\partial^2 \bar{\mathcal{L}}}{\partial q(z|x,y)^2} > 0$. Hence $\bar{\mathcal{L}}$ is convex in $q(z|x, y)$. Then the optimum can be obtained by setting the first derivative to zero. We have

$$p(x, y)\left[1 + \delta_{x,y} + \log\left(\frac{q(z|x, y)}{r(z|x)s(z|y)}\right)\right] = 0, \tag{21}$$

or equivalently

$$q(z|x, y) = \left(\frac{1}{2}\right)^{1+\delta_{x,y}} r(z|x)s(z|y). \tag{22}$$

Note that if we define

$$F(x, y) := \sum_z r(z|x)s(z|y),$$

since $\sum_z q(z|x, y) = \left(\frac{1}{2}\right)^{1+\delta_{x,y}} \sum_z r(z|x)s(z|y) = 1$, we can write

$$q(z|x, y) = \frac{1}{F(x, y)}r(z|x)s(z|y). \tag{23}$$

This is exactly the same as the second update of LatentSearch (Algorithm 1) if $r(z|x) = q(z|x), s(z|y) = q(z|y)$.

Therefore, if $q_i(z|x, y)$ is the current conditional at iteration $i$, the next update of LatentSearch (Algorithm 1) is equivalent to first solving the inner minimization of (15) thereby assigning $r(z|x) =$

$q_i(z|x), s(z|y) = q_i(z|y)$, then switching the order of the minimization operations, and solving the inner minimization of (20), therefore assigning $q_{i+1}(z|x, y) = \frac{1}{F(x,y)} q_i(z|x) q_i(z|y)$. In each of this two-step optimization iteration, either loss function goes down, or it does not change. If it does not change, the algorithm has converged. Otherwise, it cannot go down indefinitely since loss (2) is lower bounded as $I(X; Y|Z) \geq 0$ and $H(Z) \geq 0$ and therefore has to converge. This proves convergence of the algorithm to either a local minimum or a saddle point. The converged point cannot be a local maximum since it is arrived at after a minimization step. $\square$

### 7.5 Proof of Theorem 2

We first show the result for distinguishing latent graph from the triangle graph.

Since $X, Y$ are discrete variables, we can represent the joint distribution of $X, Y$ in matrix form. Let $\mathbf{M} = [p(x, y)]_{(x,y) \in [m] \times [n]}$. With a slight abuse of notation, let $\mathbf{z} := [z_1, z_2, \ldots z_k]$ be the probability mass (row) vector of variable $Z$, i.e., $\mathbb{P}[Z = i] = \mathbf{z}[i] = z_i$. Similarly, let $\mathbf{x}_z := [x_{z,1}, x_{z,2}, \ldots x_{z,k}]$ be the conditional probability mass vector of $X$ conditioned on $Z = z$, i.e., $\mathbb{P}[X = i|Z = z] = \mathbf{x}_z[i] = x_{z,i}$. Finally, let $\mathbf{y}_{z,x} := [y_{z,x,1}, y_{z,x,2}, \ldots y_{z,x,n}]$ be the conditional probability mass vector of $Y$ conditioned on $X = x$ and $Z = z$. We can write the matrix $\mathbf{M}$ as follows:

$$\mathbf{M} = \sum_{i=1}^{k} z_i \begin{bmatrix} x_{i,1}\mathbf{y}_{i,1} \\ x_{i,2}\mathbf{y}_{i,2} \\ \vdots \\ x_{i,m}\mathbf{y}_{i,m} \end{bmatrix} \tag{24}$$

Now suppose for the sake of contradiction that there exists such a $q(x, y, z)$ such that $\sum_z q(x, y, z) = p(x, y)$ and $X \perp\!\!\!\perp Y | Z$. Then $\mathbf{M}$ admits a factorization of the form

$$\mathbf{M} = \sum_{i=1}^{k} z_i' \begin{bmatrix} x_{i,1}'\mathbf{y}_{i,1}' \\ x_{i,2}'\mathbf{y}_{i,2}' \\ \vdots \\ x_{i,m}'\mathbf{y}_{i,m}' \end{bmatrix}, \tag{25}$$

where $x_{i,j}', \mathbf{y}_{i,j}', z_i'$ are due to the joint $q(x, y, z)$ and are potentially different form their counterparts in (24). Notice that since $X \perp\!\!\!\perp Y | Z$, we have $\mathbf{y}_{i,j}' = \mathbf{y}_{i,l}', \forall (j, l) \in [k] \times [m]$. Therefore the matrices

$$\begin{bmatrix} x_{i,1}'\mathbf{y}_{i,1}' \\ x_{i,2}'\mathbf{y}_{i,2}' \\ \vdots \\ x_{i,m}'\mathbf{y}_{i,m}' \end{bmatrix}, \tag{26}$$

are rank 1 $\forall i \in [k]$. Therefore, $\mathbf{M}$ has NMF rank at most $k$. Since matrix rank is upper bounded by the NMF rank, $\text{rank}(M) \leq k$. Therefore, there exists a $q(x, y, z)$ such that $\sum_z q(x, y, z) = p(x, y)$ and $X \perp\!\!\!\perp Y | Z$ *only if* $\text{rank}(M) \leq k$. In fact, it is easy to show that this is an if and only if relation: Any NMF of the joint distribution corresponds to a latent confounder and and latent confounder corresponds to an NMF of the joint distribution. Next, we show that under the generative model described in the theorem statement, this happens with probability zero.

We have the following lemma:

**Lemma 1.** *Let $\{\mathbf{x}_i : i \in [n]\}$ be a set of vectors sampled independently, uniformly randomly from the simplex $S_{n-1}$ in $n$ dimensions. Then, $\{\mathbf{x}_i : i \in [n]\}$ are linearly independent with probability 1.*

*Proof.* If $\mathbf{x}_i$ are linearly dependent, then there exists a set $\{\alpha_i : i \in [n]\}$ such that $\sum_{i=1}^{n} \alpha_i \mathbf{x}_i = 0$. Let $j = \arg\max\{i \in [n] : \alpha_i > 0\}$. Equivalently $\mathbf{x}_j$ is in the range of the set of vectors $\{\mathbf{x}_i : i \in [j-1]\}$. Therefore, we can write

$$\mathbb{P}[\{\mathbf{x}_i : i \in [n]\} \text{ are linearly independent}]$$

$$\leq \sum_{i=2}^{n} \mathbb{P}[\mathbf{x}_i \in R(\mathbf{x}_1, \ldots, \mathbf{x}_{i-1})], \tag{27}$$

where $R(\mathbf{x}_1, \ldots, \mathbf{x}_{i-1})$, is the range of the vectors $\mathbf{x}_1, \ldots, \mathbf{x}_{i-1}$, i.e., the vector space spanned by $\mathbf{x}_1, \ldots, \mathbf{x}_{i-1}$.

Notice that $\dim(R(\mathbf{x}_1, \ldots, \mathbf{x}_{i-1})) < n-1, \forall i \leq n-1$. Therefore, codimension of $R(\mathbf{x}_1, \ldots, \mathbf{x}_{i-1})$ with respect to the simplex is non-zero $\forall i \leq n-1$. Therefore, the Lebesgue measure of $R(\mathbf{x}_1, \ldots, \mathbf{x}_{i-1}) \cap S_{n-1}$ is zero with respect to the uniform measure over $S_{n-1}$. Hence, $\mathbb{P}[\mathbf{x}_i \in R(\mathbf{x}_1, \ldots, \mathbf{x}_{i-1})] = 0, \forall i \leq n-1$.

The above argument does not hold for the last term in the summation in (27). However, intersection of any $n-1$ dimensional vector space with the simplex $S_{n-1}$ is an $n-2$ dimensional slice of the simplex [51]. Therefore, it has Lebesgue measure zero with respect to the uniform measure over the simplex. $\qquad\square$

**Corollary 3.** *Let $\{\mathbf{x}_i : i \in [n]\}$ be a set of vectors sampled independently, uniformly randomly from the simplex $S_{n-1}$ in $n$ dimensions. Let $\{c_i \neq 0 : i \in [n]\}$ be arbitrary real scalars that are non-zero. Then, $\{c_i \mathbf{x}_i : i \in [n]\}$ are linearly independent with probability 1.*

*Proof.* The proof of Lemma 1 goes through since the span of a set of vectors does not change with scaling of the vectors. $\qquad\square$

$\mathbf{M}$ is rank deficient if and only if its determinant is zero, i.e., $\det(M) = 0$. The determinant is a polynomial in $\{z_i : i \in [k]\}$. By induction, one can show that if a finite degree multivariate polynomial is not identically zero, the set of roots has zero Lebesgue measure (for example, see [7]). The uniform measure over the simplex is absolutely continuous with respect to Lebesgue measure. Hence, the set of roots of a finite degree multivariate polynomial has measure zero with respect to the uniform measure over the simplex.

To show that $\det(\mathbf{M})$ is not identically zero, it is sufficient to choose a set of $z_i's$ for which determinant is non-zero. First, observe that by Corollary 3, each matrix

$$\begin{bmatrix} x_{i,1}\mathbf{y}_{i,1} \\ x_{i,2}\mathbf{y}_{i,2} \\ \vdots \\ x_{i,m}\mathbf{y}_{i,m} \end{bmatrix} \tag{28}$$

is full rank with probability 1. Let $z_1 = 1$ and $z_j, \forall j \in \{2, 3, \ldots, k\}$. Then $\det(\mathbf{M}) \neq 0$ since $\mathbf{M}$ is full rank. Therefore, the determinant, which is a polynomial in $\{z_i : i \in [k]\}$ is not identically zero. This concludes the proof that with probability 1, $\text{rank}(\mathbf{M}) = n > k$.

If the distribution is generated from the direct graph, from Lemma 1, we know that the rows of conditional probability matrix are linearly independent. Since joint probability matrix can be obtained by scaling each row of this matrix with the probability values of $X$, and this operation does not change rank, joint probability matrix obtained from the direct graph is full rank with probability 1. Therefore non-negative rank of this matrix has to be $n$, concluding the proof. $\qquad\square$

## 7.6 Proof of Corollary 1

The statement follows from the fact that the proposed generative model induces a non-zero probability measure on every joint distribution, which is the set of distributions that can be encoded from the *triangle graph* and any distribution that can be encoded by the *latent graph* requires $X \perp\!\!\!\perp Y \,|\, Z$, which we show in Theorem 2 happens with probability zero. $\qquad\square$

## 7.7 Proof of Theorem 3

We give the proof for binary $Z$. The argument can be extended to when $Z$ has any finite number of states.

We overload the notation and use $z$ for the probability that random variable Z is 0. We have

$$p(Z = 0) = z, \qquad p(Z = 1) = 1 - z \tag{29}$$
$$p(X = 0|Z = z) = x_z \tag{30}$$
$$p(Y = 0|X = x, Z = z) = y_{x,z} \tag{31}$$

The conditional distributions from UGM can be sampled uniformly from the simplex via normalized exponential random variables, however in the case of binary variables, this is equivalent to sampling uniformly. Hence, we can assume $x_z, y_{x,z}$ are uniform random variables with support $[0, 1]$. Based on this generative model, we can calculate $p(x)$ and $p(y|x)$ as follows:

$$p(X = 0) = x_0 z + x_1 (1 - z), \tag{32}$$
$$p(X = 1) = (1 - x_0)z + (1 - x_1)(1 - z)$$

$$p(Y = 0|X = 0) = \frac{y_{0,0} x_0 z + y_{0,1} x_1 (1 - z)}{x_0 z + x_1 (1 - z)} \tag{33}$$

$$p(Y = 0|X = 1) = \frac{y_{1,0}(1 - x_0)z + y_{1,1}(1 - x_1)(1 - z)}{(1 - x_0)z + (1 - x_1)(1 - z)} \tag{34}$$

We use the characterization of [31] for the minimum entropy $Z$ that can make $X, Y$ conditionally independent. Let $t = p(X = 0)$ and let $\alpha := p(Y = 0|X = 0), \beta := p(Y = 0|X = 1)$. We re-state their theorem for self-containment of our paper:

**Theorem 4** ([31]). *Consider two binary random variables $X, Y$. Define $t := p(X = 0), \alpha := p(Y = 0|X = 0), \beta := p(Y = 0|X = 1)$. Let $\alpha' = \min\{\alpha, \beta\}, \beta' := \max\{\alpha, \beta\}$. Then of all $q(x, y, z')$ where $q(x, y) = p(x, y)$ and $X \perp\!\!\!\perp Y |Z$ minimum entropy $Z'$ has entropy*

$$LB := \min\{H_b(A), H_b(B)\}, \tag{35}$$

$$A = t\left(1 - \frac{\alpha'}{\beta'}\right), B = (1 - t)\left(1 - \frac{1 - \beta'}{1 - \alpha'}\right) \tag{36}$$

Note that in the generative model we are considering, the entries of $p(x, y)$ are random variables, which implies that $LB$ is a random variable.

Consider a sequence $z_n$. Let $Z_n$ be the binary random variable where $\mathbb{P}(Z_n = 0) = z_n$. Notice that $H(z_n)$ converges to zero if and only if $z_n$ converges to either 0 or 1. Since the generative model is symmetric with respect to the conditionals $p(x|z = 0)$ compared to $p(x|z = 1)$ and $p(y|x, z = 0)$ compared to $p(y|x, z = 1)$, without loss of generality we can consider the case where $z_n$ goes to 0.

Now suppose $0 < z_0 < 0.5$ and $z_n$ is a monotonically decreasing sequence. When we substitute $z_n$ for $z$ in the generative model, we use the symbols in Theorem 4 with subscript $n$ to distinguish them for different values of $n$.

The event that there does not exist a latent variable with small entropy that can make the observed variables independent is equivalent to the event that the lower bound is strictly greater than the entropy of the true latent variable:

$$\mathbb{P}(\mathcal{Q}_p = \emptyset) = \mathbb{P}(p(x, y) : \nexists q(x, y, z') \tag{37}$$

$$\text{s.t. } \sum_{z'} q(x, y, z') = p(x, y), \tag{38}$$

$$X \perp\!\!\!\perp Y |Z', H(Z') \le H(Z)) \tag{39}$$
$$= \mathbb{P}(LB > H(Z)) \tag{40}$$

We want to show that

$$\lim_{n \to \infty} \mathbb{P}(LB_n > H(Z_n)) = 1. \tag{41}$$

Define the following events:

$$\varepsilon_{z_n}^A := \{\text{Event that } H_b(A_n) \le H_b(z_n)\}. \tag{42}$$
$$\varepsilon_{z_n}^B := \{\text{Event that } H_b(B_n) \le H_b(z_n)\}. \tag{43}$$

By union bound

$$\mathbb{P}(LB_n \le H(Z_n)) \le \mathbb{P}(\varepsilon_{z_n}^A) + \mathbb{P}(\varepsilon_{z_n}^B) \tag{44}$$

We first investigate the term $\lim_{n \to \infty} \mathbb{P}(\varepsilon_z^A)$. By conditioning on the event that $A_n \le 0.5$ and $A_n > 0.5$, we can reduce the comparison between $H_b(a), H_b(c)$ to a comparision between $a$ and $c$.

Due to first applying the law of total probability and then Bayes rule, we have

$$\mathbb{P}(\varepsilon_{z_n}^A) = \mathbb{P}\left(\varepsilon_{z_n}^A | A_n \leq 0.5\right)\mathbb{P}(A_n \leq 0.5) + \mathbb{P}\left(\varepsilon_{z_n}^A | A_n > 0.5\right)\mathbb{P}(A_n > 0.5)$$
$$= \mathbb{P}\left(A_n \leq z_n | A_n \leq 0.5\right)\mathbb{P}(A_n \leq 0.5) + \mathbb{P}\left(A_n \geq 1 - z_n | A_n > 0.5\right)\mathbb{P}(A_n > 0.5)$$
$$= \mathbb{P}\left(A_n \leq z_n\right)\mathbb{P}(A_n \leq 0.5 | A_n \leq z_n) + \mathbb{P}\left(A_n \geq 1 - z_n\right)\mathbb{P}(A_n > 0.5 | A_n \geq 1 - z_n)$$
$$= \mathbb{P}\left(A_n \leq z_n\right) + \mathbb{P}\left(A_n \geq 1 - z_n\right)$$

Define the following random variable:

$$S_n^A := -z_n + t_n\left(1 - \frac{\alpha_n'}{\beta_n'}\right), \tag{45}$$

where the terms on the right hand side are as defined in Theorem 4. Then $\mathbb{P}\left(A_n \leq z_n\right) = \mathbb{P}\left(S_n^A \leq 0\right)$ and $\mathbb{P}\left(A_n \geq 1 - z_n\right) = \mathbb{P}\left(S_n^A \geq 1\right)$. We have

$$\mathbb{P}\left(S_n^A \leq 0\right) = \int_{-\infty}^0 S_n^A d\mu, \tag{46}$$

where $\mu$ is the probability measure induced by the generative model. Note that $t_n \in [0, 1], z_n \in (0, 0.5), \frac{\alpha_n'}{\beta_n'} \in (0, 1]$, we have $|S_n| \leq 1$. Then from the dominated convergence theorem since $\int 1 d\mu = 1 < \infty$, we have

$$\lim_{n \to \infty} \int_{-\infty}^0 S_n^A d\mu = \int_{-\infty}^0 \lim_{n \to \infty} S_n^A d\mu. \tag{47}$$

We have

$$\lim_{n \to \infty} S_n^A = \lim_{n \to \infty} -z_n + t_n\left(1 - \frac{\alpha_n'}{\beta_n'}\right) \tag{48}$$

Both $\alpha_n$ and $\beta_n$ are random variables supported on $[0, 1]$. Moreover, since limit exists for $\alpha_n, \beta_n$, it also exists for $\alpha_n' := \min\{\alpha_n, \beta_n\}$, similarly it exists for $\beta_n'$. Therefore,

$$\lim_{n \to \infty} -z_n + t_n(1 - \frac{\alpha_n'}{\beta_n'}) = x_1\left(1 - \frac{\lim_n \alpha_n'}{\lim_n \beta_n'}\right) \tag{49}$$

$$= x_1\left(1 - \frac{\lim_n \min\{\alpha_n, \beta_n\}}{\lim_n \max\{\alpha_n, \beta_n\}}\right) \tag{50}$$

$$= x_1\left(1 - \frac{\min\{\lim_n \alpha_n, \lim_n \beta_n\}}{\max\{\lim_n \alpha_n, \lim_n \beta_n\}}\right) \tag{51}$$

$$= x_1\left(1 - \frac{\min\{y_{0,1}, y_{1,1}\}}{\max\{y_{0,1}, y_{1,1}\}}\right) \tag{52}$$

where the last equation follows from the equations (32)-(34). Finally, we have that

$$\int_{-\infty}^0 x_1\left(1 - \frac{\min\{y_{0,1}, y_{1,1}\}}{\max\{y_{0,1}, y_{1,1}\}}\right) d\mu \tag{53}$$

$$= \mathbb{P}\left(x_1\left(1 - \frac{\min\{y_{0,1}, y_{1,1}\}}{\max\{y_{0,1}, y_{1,1}\}}\right) \leq 0\right) \tag{54}$$

$$= \mathbb{P}\left(x_1\left(1 - \frac{\min\{y_{0,1}, y_{1,1}\}}{\max\{y_{0,1}, y_{1,1}\}}\right) = 0\right) = 0 \tag{55}$$

$$\tag{56}$$

where the last two equations follow from the fact that $x_1\left(1 - \frac{\min\{y_{0,1}, y_{1,1}\}}{\max\{y_{0,1}, y_{1,1}\}}\right)$ is supported in the interval $[0, 1]$ and has an absolutely continuous measure, which implies that measure of a single point is zero.

Similarly, we can calculate

$$\mathbb{P}\left(S_n^A \geq 1\right) = \int_1^\infty S_n^A d\mu = 0, \tag{57}$$

which leads to $\mathbb{P}\left(\varepsilon_{z_n}^A\right) = 0$.

Next, we consider the same analysis for $\mathbb{P}\left(\varepsilon_{z_n}^B\right)$. One difference is the replacement of $t_n$ with $1 - t_n$ which does not affect the derivation except for the replacement of $x_1$ with $1 - x_1$. Moreover, in the numerator within the paranthesis in (55, $\min\{y_0, 1, y_{1,1}\}$ is replaced with $1 - \max\{y_0, 1, y_{1,1}\}$ and similarly in the denominator $\max\{y_0, 1, y_{1,1}\}$ is replaced with $1 - \min\{y_0, 1, y_{1,1}\}$. It follows that $\mathbb{P}\left(\varepsilon_{z_n}^B\right) = 0$. This implies that $\lim_{n \to \infty} \mathbb{P}\left(LB_n \leq H(Z_n)\right) = 0$ concluding the proof.

$\square$

### 7.7.1 Comments on Distinguishing Direct Graph from Latent Graph with Entropy

Consider the uniform generative model for the triangle graph. It is easy to see that in this case, in (36), $t, \alpha', \beta'$ become independent and uniformly distributed random variables with the compact support $[0, 1]$. One can calculate the distribution of this lower bound accordingly. This can be used to obtain the probability of identifiability between direct graph and the latent graph for a given upper bound on the entropy of the latent variable. We do not pursue this calculation here.

### 7.7.2 Extension to $Z$ with $k$ states

Consider the setting where $Z$ has $k$ states. We use the following notation in this section:

$$p(Z = i) = z^{(i)}. \tag{58}$$

First, note that the characterization of [31] is still applicable since they show increasing the dimension of $Z$ to more than two states cannot reduce the minimum entropy. Similar to the above proof, we will assume a sequence of random variables $Z_n$. Let $Z_n$ be a sequence of random variables with the pmf

$$p(Z_n = i) = z_n^{(i)}. \tag{59}$$

Note that $H(Z_n) \to 0$ if and only if $\exists i \in [k]$ such that $z_n^{(i)} \to 1$ and $(z_n^{(j)})_{j \neq i} \to \mathbf{0}$. In the following, we show that a similar analysis to the binary case goes through irrrespective of how $(z_n^{(j)})_{j \neq i}$ converges to the zero vector. Suppose without loss of generality $z_n^{(1)} \to 1$.

Due to the grouping rule of entropy, we have

$$H(Z_n) = H_b(z_n^{(1)}) + (1 - z_n^{(1)})H((w_n^i)_{2 \leq i \leq k}), \tag{60}$$

where $w_n^{(i)} = \frac{z_n^{(i)}}{1 - z_n^{(1)}}$. Let $N$ be such that $z_N^{(1)} \geq 1 - \frac{\epsilon}{\log_2(k)}$. Then $z_n^{(1)} \geq 1 - \frac{\epsilon}{\log_2(k)}, \forall n > N$. Then we have $H(Z_n) \leq H_b(z_n^{(1)}) + \epsilon, \forall n \geq N$.

Now we can replicate the proof for the binary $Z$ as follows. Let us define the events:

$$\varepsilon_n^A := \{\text{Event that } H_b(A_n) \leq H_b(z_n^{(1)})\}.$$
$$\varepsilon_n^B := \{\text{Event that } H_b(B_n) \leq H_b(z_n^{(1)})\}.$$
$$\delta_n^A := \{\text{Event that } H_b(z_n^{(1)}) < H_b(A_n) \leq H(Z_n)\}.$$
$$\delta_n^B := \{\text{Event that } H_b(z_n^{(1)}) < H_b(B_n) \leq H(Z_n)\}.$$

By union bound

$$\mathbb{P}(LB_n \leq H(Z_n)) \leq \mathbb{P}(\varepsilon_n^A) + \mathbb{P}(\varepsilon_n^B) + \mathbb{P}(\delta_n^A) + \mathbb{P}(\delta_n^B) \tag{61}$$

We can write

$$\mathbb{P}(\delta_n^A) \leq \int_{H_b(z_n^{(1)})}^{H_b(z_n^{(1)}) + \epsilon} H_b(A) d\mu. \tag{62}$$

It is easy to see that $\lim_{n \to \infty} \mathbb{P}(\delta_n^A) = 0$ since $\epsilon \to 0$ and the measaure induced by the generative model is absolutely continuous in the integral.

The rest of the analysis follows similarly to the binary case: We can obtain expressions for $\alpha, \beta$ using $k$ terms instead of 2. In the limit, all but the term that contains $z_n^{(1)}$ go to zero and we can conclude the proof using the same arguments. $\square$

Figure 5: The measure of distributions from the triangle graph that does not fit to latent graph with a latent at least as simple as the true latent. As the entropy of the true latent goes to $0$, this fraction goes to $1$. This is precisely the measure of models which are identifiable with our entropic approach in the case of binary $X, Y, Z$.

Figure 6: Fraction of distributions from the triangle graph that does not fit to latent graph with a latent at least as simple as the true latent for binary $X, Y$ and ternary $Z$. As the entropy of the true latent goes to $0$, this fraction goes to $1$. This is precisely the fraction of models which are identifiable with our entropic approach in the case of binary $X, Y, Z$.

### 7.8 Entropic Identifiability for Binary Variables

For $H(Z) > 0$, we can approximate the fraction of identifiable causal models via simulations. We sample probability distributions from the uniform generative model. For each sample we check if the entropy of the true latent $H(Z)$ is strictly less than the common entropy of observed variables $G_1(X, Y)$. These are the models from triangle graph which cannot be fit onto latent graph with a low-entropy latent. Figure 5 shows this fraction. $\sigma_c$ is the parameter of the Dirichlet distribution used to sample the conditional distributions from the Triangle graph. $\sigma_c = 1$ corresponds to uniform sampling model and smaller the $\sigma_c$ more deterministic the conditional distributions are in the sampling model.

See Figure 6 for the corresponding plot for ternary $Z$.

### 7.9 $I(X; Y | Z)$ vs. $H(Z)$ Tradeoff Curve

Figure 7 shows the $I(X; Y | Z)$-$H(Z)$ tradeoff LatentSearch (Algorithm 1) obtains for a joint distribution sampled as follows: The distribution of $Z$ as well as the conditional distributions $p(X | z), p(Y | z), \forall z$ are chosen uniformly at random over the simplex.

Figure 7: $I(X;Y|Z)$ vs. $H(Z)$ tradeoff curve obtained by LatentSearch (Algorithm 1) for an arbitrary joint $p(x,y)$ from the graph $X \leftarrow Z \rightarrow Y$. We observed that the curve's shape is consistent across many runs irrespective of the graph, although the crossing point where $I(X;Y|Z) = 0$ changes.

## 7.10 Complete Details of Experiments

In this section, we explain some of the key implementation details for the experiments in Section 5 that were left out from the main text due to space constraints.

**Sampling a low-entropy latent:** In many experiments, we sample distributions from either the latent graph or the triangle graph such that entropy of the latent confounder $Z$ is small. For example, we enforce $H(Z) \le \theta$ for varying thresholds $\theta$ in Figure 2c . We use a form of rejection sampling combined with sampling from Dirichlet distributions with low-entropy as follows:

Suppose, we need $N$ samples where $H(Z) \le \theta$. We initialize $\alpha^{(0)} = 1$ and obtain $10N$ samples from $Dir(\alpha^{(0)})$. If we have at least $N$ samples where $H(Z) \le \theta$, we are done. If not, we update $\alpha$ by halving it, i.e., $\alpha^{(1)} = 0.5\alpha^{(0)}$. The lower the $\alpha$ value, the lower-entropy distributions we will obtain from $Dir(\alpha)$. Then we repeat this process until iteration $i$, where at least $N$ samples can be obtained from $10N$ samples using $Dir(\alpha^{(i)})$. We conclude by analyzing the histogram plots of $H(Z)$ that this method not only allows us to sample distributions where $H(Z) \le \theta$ but also where $H(Z) \approx \theta$, providing us with a better control over the entropy of the latent confounder.

**Choosing number of states of latent variable in LatentSearch:** Recall that LatentSearch allows us to discover a tradeoff between $H(Z)$ and $I(X;Y|Z)$, which, combined with a $I(X;Y|Z)$ for conditional independence, can be used to approximate common entropy. Since only $X, Y$ are observed, we do not know how many states $k$ $Z$ has. As pointed out in the main text, one can try all $k \le mn$ without loss of generality, where $m, n$ are the number of states of $X, Y$, respectively. However, in practice, this takes a long time. Furthermore, we identified that this is not necessary for the estimation of common entropy.

We observe that if we search over $Z$ with very large number of states, e.g., $k = mn$, performance of LatentSearch does not improve compared to having $k = \min\{m, n\}$. This is because the number of optimization parameters increases significantly which may require many more iterations. It also slows down the algorithm. We observed that choosing $k = \max\{m, n\}$ provides the smallest entropy latents in practice. Therefore, we set $k = \max\{m, n\}$ in LatentSearch for our experiments.

**Sampling DAGs for testing EntropicPC in Figure 4:** We first sample Erdös-Rényi graphs with parameter 0.2. Since there are 10 nodes, this corresponds to an average degree of 2 per node. Note that these graphs are undirected. We need to make them directed and ensure there are no cycles. For this, we randomly picked a total order between the nodes and directed the edges respecting that total order. It can be easily shown that the resulting graphs have no cycles.

**Sampling joint distributions for a given DAG in Figure 4:** For every DAG we generate, we need to obtain a joint distribution from which we sample a dataset. To obtain a distribution for a given graph, we employ a method from Chickering and Meek [10]. It is known that constraint-based methods require the data to be faithful to the graph, i.e., every pair of variables that are connected in the graph should be dependent. This notion should also be true under any conditioning set. In practice, this does not always hold. Specifically, nodes that are far away from each other in the graph might be almost statistically independent. To ensure faithfulness in practice, Chickering and Meek

use a method to sample conditional distributions for a given DAG [10]. In summary, it ensures that parent-child relations are far from uniform. The details of their sampling method, which we also use, are as follows:

For a variable $X$ with $m$ states, they define the vector $\mathbf{v} = \frac{1}{T}(\frac{1}{1}, \frac{1}{2}, \dots, \frac{1}{m}), T = \sum_{i=1}^{m} \frac{1}{i}$. For the $j^{th}$ instantiation of the parent set of $X$, $pa_x^{(j)}$, they use the vector $\mathbf{v}_j$ which is the $j$-shifted version of $\mathbf{v}$. The shifts are cyclic in the sense that $\mathbf{v}_m = \mathbf{v}$. They later sample $p(X|pa_x^{(j)})$ from the Dirichlet distribution with parameter vector $\mathbf{v}_j$. When each coordinate parameter of Dirichlet is identical, the expected distribution is uniform. When Dirichlet is sampled with parameters $\mathbf{v}_j$ as given above, each coordinate has a different parameter. Indeed, the expected distribution becomes $\mathbf{v}_j$ rather than uniform. Therefore, this method ensures that $p(x|pa_x^j)$ is typically far from uniform, encouraging strong dependence between parents and the children in the graph.

**Details specific to Figure 2a:** We sampled 100 distributions from the latent graph for each value of $n$, where $X, Y, Z$ all have $n$ states. In each distribution, we ensure that $H(Z) \leq 1$ using the low-entropy sampling method described above. We use LatentSearch on 50 different values of $\beta$, uniformly spaced in the interval $[0, 0.1]$. We set the latent variable for LatentSearch to $n$ number of states. We run LatentSearch for 500 iterations each time. We used the conditional mutual information threshold of 0.001: In other words, of the algorithm outputs for the 50 $\beta$ values used, we pick the smallest entropy $Z$ discovered by the algorithm among those that ensure $I(X; Y|Z) \leq 0.001$. We then compare this value with the entropy of the true latent confounder.

**Details specific to Figure 2b:** We sample 1000 distributions from the triangle graph. As mentioned in the main text, we use LatentSearch output to approximate common entropy. The settings for LatentSearch are the same as above, i.e., we use 50 $\beta$ values uniformly spaced in the range $[0, 0.1]$, use $n$ states for the latent and run the algorithm for 500 iterations. Entropy recovered by LatentSearch for a pair $X, Y$ is then compared with $\min\{H(X), H(Y)\}$. The $y-$axis shows that fraction of times the reconstructed latent has entropy of at least $\alpha \min\{H(X), H(Y)$ for different values of $\alpha$.

**Details specific to Figure 2c:** For this figure we sample 1000 distributions from both triangle graph and the latent graph for various upper bounds on the entropy of $Z$. Low-entropy sampling is done as explained before. Finally, Algorithm 2 is used to identify the true causal graph with $\theta = 0.8 \min\{H(X), H(Y)\}$. LatentSearch settings used within Algorithm 2 are as given previously.

**Details specific to Figure 3a:** Tuebingen dataset consists of around 100 real cause-effect pairs. We run LatentSearch to understand whether real cause-effect pairs can be made independent by low-entropy variables. As explained in the main text, we used different conditional independence thresholds. Visual inspection of $I - H$ curves suggest that 0.001 is a good threshold for this dataset. This can be done by checking, for the given range of $\beta$ values, where the curve disengages form the $x = 0$ axis. We used 100 $\beta$ values in the range $[0, 0.1]$ and run the LatentSearch algorithm for 1000 iterations for this experiment.

**Details specific to Figure 3b:** This figure is an example of the tradeoff curve LatentSearch discovers for various values of $\beta$. Each dot corresponds to a joint distribution $p(x, y, z)$ constructed by LatentSearch for a given value of $\beta$ after a certain number of iterations. As can be seen from (2), smaller $\beta$ values enforce smaller $I(X; Y|Z)$. The horizontal line indicates $\min\{H(X), H(Y)\}$. $X, Y$ can always be separated with this much entropy since by definition $X \perp\!\!\!\perp Y | X$ and $X \perp\!\!\!\perp Y | Y$. Ideally, i.e., with infinite samples and infinitely many $\beta$ values, the point that intersects the $x = 0$ line (i.e., the $y-$ axis) should give the common entropy. To account for finite-sample effects, we use a different horizontal line, which we call conditional mutual information threshold, as described before.

**Details specific to Figure 3c:** We sample from the graph $X \to Z \to Y$ and investigate $H(Z)$. Note that $Z$ acts as a mediator if it is not observed. Our goal is to understand if it is typical to have low-entropy mediators. We set the dimensions of $X, Y, Z$ to $n$. If $Z$ has $k$ states, $H(Z) \leq \log(k)$. Our goal is to demonstrate that unless $k$ is a constant, $H(Z)$ scales similar to $H(X), H(Y)$. Most of the details of this experiment are provided in the main text. Note that when $\alpha_{Dir} \leq \frac{1}{n}$ for a distribution with $n$ states, a sample from Dirichlet distribution typically looks very peaked, i.e., it has very high probability for one of the states, and very low probabilities for the rest. When such a low $\alpha_{Dir}$ is used to sample the conditional of $p(Z|x)$ for every value of $x$, $X$ and $Z$ are almost deterministically related, i.e., there is almost no additional entropy introduced in the system. We show that even then the entropy of the mediator scales. Larger $\alpha_{Dir}$ values will give distributions that are closer to uniform, which in turn will make $Z$ close to uniform and have $\log(n)$ entropy.

(a)                                                    (b)

Figure 8: [Section 5.5] Output of **(a)** EntropicPC and **(b)** PC in ADULT Data.

**Details specific to Figure 4:** Our goal is to demonstrate how well output of EntropicPC approximates a true causal graph by checking graphical distances between the skeleton and essential graph. Essential graph is the mixed graph where undirected edges show the edges that cannot be learned whereas directed edges show the edges that can be learned. Structural Hamming Distance counts the number of edges that should be reverted, added or removed to change the output graph into the true graph. For skeleton discovery, we see edge discovery as a classification problem and calculate the F-score as an established summary of the classifier performance. The graph and distribution sampling are described above. We sample a dataset with $100000$ variables and for each figure, we subsample varying number of samples from this dataset without replacement. This is repeated for $100$ different graphs, and their corresponding distributions.

## 7.11   EntropicPC and PC on ADULT Dataset

Due to space constraints in the main text, we provide the outputs of PC and EntropicPC algorithms for the ADULT dataset in this section. The results are given in Figure 8.

Note that the bidirected edges represent undirected, i.e., unoriented edges. Even though the true causal graph is not known, we can easily conclude that EntropicPC discovers a much more reasonable graph: *salary* is caused by *education,occupation* whereas PC misses both edges. Both algorithms seems to suffer from unfaithful data - *sex* is not required to separate *marital-status* and *occupation* whereas we expect it to since it should be a source node. This drives both algorithms to orient *sex* as a collider.

## 7.12   EntropicPC on Line and Collider Graphs

To demonstrate effectiveness of EntropicPC compared to PC on the simplest possible graph, we conducted the experiments of Figure 4 on the line graph $X \rightarrow Y \rightarrow Z$ and the collider graph $X \rightarrow Y \leftarrow Z$. The results are given in Figures 9 and 10, respectively.

## 7.13   Comparing LatentSearch with EM, NMF and Gradient descent

### 7.13.1   Comparison to gradient descent

Instead of using LatentSearch for minimizing the loss in $(2)$, one can use gradient descent. Even though the objective is not convex, gradient descend will still output a stationary point if it converges. However gradient descend comes with many practical issues, as we detail in the following, and support with experiments.

Figure 9: Performance of EntropicPC, EntropicPC-C and PC in the graph $X \to Y \to Z$. All three methods perform identically for $10000$ samples (not shown). Both EntropicPC and EntropicPC-C consistently outperforms baseline PC.

First, we observed that iterative update step is slightly faster than the gradient descent step: Average time for iterative update:0.000063 seconds. Average time for gradient update: 0.000078 seconds. More importantly, gradient descent takes much longer to converge and does not even achieve the same performance.

As observed in Figure 11, gradient descent converges only after 350000 iterations, whereas we observed that iterative update converges after around 200 iterations. Based on the average update times, this corresponds to a staggering difference of 0.01 seconds for the iterative algorithm vs. 27.3 seconds for the gradient descent algorithm. Although these results are for when $n = m = k = 5$ states, we observed single iterative update to be faster than single gradient update, giving similar performance comparison results for $n = m = k = 80$ states.

The above result is for a constant step size of $0.001$. With smaller step size, convergence slows down even further. With larger step size, gradient descent does not converge.

### 7.13.2 Comparison with EM algorithm

EM is the first algorithm suggested for solving the pLSA problem [19]. For the details of EM within this framework, please see [19]. However the EM algorithm for pLSA problem does not have any incentive to minimize the entropy of the latent factor.

In order to see how EM affects the entropy of the discovered latent variable, we run EM algorithm by initializing it at the points that are output by LatentSearch (Algorithm 1). Results are illustrated in Figure 12. We observe that the points obtained in the $I - H$ plane migrate towards $I(X; Y|Z) = 0$ line, while staying above what we believe is a fundamental lower bound curve. This step does not improve our algorithm, as it increases the entropy of the latent variables.

### 7.13.3 Comparison with NMF

Consider the joint distribution matrix $\mathbf{M}$. Suppose we find an approximation to this matrix as $\mathbf{M} \approx \mathbf{U}\mathbf{V}$ where the common dimension of $\mathbf{U}, \mathbf{V}$ is $k$ through NMF. This is equivalent to setting the dimension of the latent variable to $k$. This can be seen as a hard entropy threshold on the entropy of the latent factor since $H(Z) \leq \log(k)$. We can sweep through different dimensions and see how NMF performs compared to LatentSearch (Algorithm 1). Note that NMF is in general hard to solve. A commonly used approach is the iterative algorithm: Initialize $\mathbf{U}_0, \mathbf{V}_0$. Find the best $\mathbf{U}_1$ such that $\mathbf{M} \approx \mathbf{U}_1\mathbf{V}_0$. Then find the best $\mathbf{V}_1$ such that $\mathbf{M} \approx \mathbf{U}_1\mathbf{V}_1$ and iterate. In the experiments, we used this iterative algorithm together with $l_1$ loss.

Figure 10: Performance of EntropicPC, EntropicPC-C and PC in the graph $X \rightarrow Y \leftarrow Z$. All three methods perform identically for 10000 samples (not shown). Different from the line graph, for collider graph in the very small sample regime, even though the proposed methods significantly outperforms PC in terms of skeleton discovery, PC gets a slight edge in performance in terms of SHD. Note that PC tends to declare variables independent in the small sample regime. It is likely that for 100 samples, PC often estimates empty graph, which has SHD of 2 from the essential graph. This explains the discrepancy between the performance for F1 and SHD scores for 100 samples.

Figure 11: Comparison of the iterative algorithm with gradient descent. Blue points show the trajectory of gradient descent, whereas orange points show the trajectory for Algorithm 1 for 10 randomly initialized points with different $\beta$ values in loss (2). Gradient descent takes 350,000 iterations to converge whereas iterative algorithm converges in about 200 iterations. Moreover, the points achieved by iterative algorithm are strictly better than gradient descent after convergence.

Figure 12: Applying EM to the output of iterative algorithm migrates points to $I(X;Y|Z=0)$ line: (a) Latent variables discovered by LatentSearch (Algorithm 1) shown on the $I(X;Y|Z) - H(Z)$ plane. (b,c) After applying EM algorithm on the points in (a) after 60 and 300 iterations. Observe that the points always remain above the line depicted by LatentSearch (Algorithm 1).

(a) Causal Graph $X \leftarrow Z \rightarrow Y$        (b) Causal Graph $X \leftarrow Z \rightarrow Y, X \rightarrow Y$

Figure 13: Comparison of the iterative algorithm to NMF for when $|X| = |Y| = 20, |Z| = 10$. When the true model comes from the causal graph $X \leftarrow Z \rightarrow Y$ in (a), iterative algorithm successfully finds latent variables that with entropy at most true latent entropy (shown as blue horizontal line), whereas NMF cannot achieve the same performance, irrespective of the dimension restriction to the latent variable. In (b) data comes from the causal model $X \leftarrow Z \rightarrow Y, X \rightarrow Y$. Although neither algorithm can identify a latent factor that makes $X, Y$ conditionally independent (vertical blue line), iterative algorithm finds strictly better latent factors in terms of both small entropy and conditional mutual information between $X, Y$.

## Footnotes

[2]For a formal introduction to Pearl's framework please see [47, 38].

[3]In some proofs, $x_i$ is used to represent the probability that the variable $X$ takes the value $i$ for simplicity.