[Reviews · NeurIPS 2020]

Review 1

Summary and Contributions: This paper considers the minimum entropy of a latent variable that renders two variables conditionally independent and explores the relevance of this quantity for causal discovery. The authors describe two ideas for employing this quantity: first they use it to distinguish cause-effect relations from confounding subject to appropriate assumptions and second as a necessary condition for separating sets in PC type algorithms.

Strengths: The theoretical claims are sound, assumptions are clearly stated. I liked the estimation of common entropy for real-world cause-effect pairs, which supported the claim of the authors that it scales with minimum marginal entropy.

Weaknesses: The main weakness is the missing contribution. The first part (telling confounding from cause-effect relation) is close to [23] (which is published at UAI 2011). The second part (necessary condition for PC type algorithms) is novel to the best of my knowledge, but its benefit remains unclear, despite some experimental evidence. The reason is that small sample problems also occur for the estimation of common entropy and there is no theoretical evidence that considering common entropy mitigates the risk of identifying the wrong separating sets.

Correctness: The approach of stating strong assumptions regarding statistics and causality, which renders causal discovery problems solvable that ares unsolvable otherwise is an interesting direction. The authors state and motivate their assumptions explicitly and correctly derive mathematical conclusions.

Clarity: I didn't see major issues with clarity, although I found the use of Renyi entropy as opposed to Shannon entropy comes with lack of motivation.

Relation to Prior Work: I didn't see a discussion of [23] which contains already one of the two main ideas of the paper. Another paper that sounds relevant to me: Bastian Steudel, Nihat Ay: Information-Theoretic Inference of Common Ancestors https://www.mdpi.com/1099-4300/17/4/2304

Reproducibility: Yes

Additional Feedback:


Review 2

Summary and Contributions: this paper applies common entropy for learning causal graphs from observational data. the paper demonstrates the use of common entropy for causal discovery with two examples. In the first example, assume that there are two observed variables, X, Y. The paper show that if the latent confounder is simple, and there is no simple mediators in between two variable, this problem can be solved using common entropy. In the second example, the authors show that common entropu can be used to improve any constraint-based causal discovery algorithm in the small sample regime.

Strengths: To the best of my knowledge, the applications of common entropy to causal discovery are original.

Weaknesses: The problem & method are not well motivated. For example, the motivation for using common entropy is to avoid the issue of high dimensional covariate X and Y. However, in the first example, the authors used simple binary variables for both X and Y.

Correctness: The empirical validation for ADULT dataset seems quite questionable. There is no ground truth in the dataset, but the authors assumed some causal relationships and validate the method based on that. I did not thoroughly check the proofs of the theorems.

Clarity: The writing can use substantial improvement. For example, in line 65, the authors wrote "this problem can be solved using common entropy", without referring what the problem is. There are also a lot of incomplete sentences, such as the caption of Figure 2.

Relation to Prior Work: The connection to prior work is not clearly examined in the main manuscript.

Reproducibility: Yes

Additional Feedback:


Review 3

Summary and Contributions: I thank the authors for addressing my concerns. I would urge the authors to include next time into their rebuttal additional experimental results when the reviewers ask for them. The rebuttal doesn't provide any additional supporting experimental results and as the experimental evaluation is the main weakness of this paper, I will not raise my score. ================ This paper examines the problem of learning the simplest latent variable that make tow discrete variables conditionally independent. It uses Renyi common entropy for this task and presents an iterative approach to estimating this. Further, it shows how this quantity can be used for causal inference in conjunction with constraint-based approaches.

Strengths: This paper addresses an interesting problem and applies their finding to causal inference which is an area of growing interest at the conference. This paper is overall well written and manages to discuss a technical topic in an accessible way, making it a pleasant read. It proposes a new class of entropy measures and provides theory for the use of these measures in causal discovery. Further, it proposes a practical algorithm for the estimation of the in general intractable measures. Based on the theory it shows how this can be combined into a practical addition to any constraint-based causal discovery method which allows it to orient more of the otherwise unoriented egdes. In summary, I believe that this paper would be of interest to the community and that it is novel enough.

Weaknesses: The main weaknesses are: * experimental evaluation * a little bit on the clarity of the writing (see below) Experiments: My main criticism is that experiments span a little over 2 pages, there is very little in terms of comparison to related work. It is very nice that the performance of the proposed iterative procedure for estimating common entropy is evaluated, but there is no comparison to any related methods. Please compare this (maybe to [30]?). A lot of the experimental section is spent in validating assumptions and conjectures. I would suggest that the authors put these experimental results in the appendix and focus on comparison of the proposed methods (latentsearch and entropicpc) to related work. For the evaluation of EntropicPC, I would strongly urge the authors to compare their methods against any of the many, many extension of the PC algorithm (and not just against PC itself). Further, I would suggest that the authors test on more real-world datasets than just the ADULT dataset.

Correctness: To the best of my knowledge the theoretical results presented in this paper are correct. To the best of my knowledge the empirical methodology used is correct.

Clarity: This paper is overall clearly written. There is a nice introduction to the examined problem and nice motivation for its usefulness in causal inference. Also, there is a nice explanation of common entropy in the introduction that sets up the rest of the paper nicely. Also, I would like to complement the authors for clearly stating the assumptions used (page 4). For the theoretical results (section 3), I would suggest to the authors that only the central results are kept in the main text and the corollaries relegated to the appendix. The main weaknesses of the writing are: * a conjoined introduction and discussion related work this does not make section 1 “flow” very nicely. i would suggest that the authors extract the discussion of related work into a separate section and put this section right after section 1 * missing references in many parts of the text — i would urge the authors to add references to all claims and findings that are not their own. Some examples: line 50 “common entropy”, line 141 Reichenbach’s common cause principle, sentence in line 203/204 Also, please add additional explanations for Figure 1 and Figure 4. Typos and suggestions: * line 92 — shown -> denoted * change subtitles in experimental section 5.1 -> LatentSearch and remove mention of Sections from subtitles

Relation to Prior Work: From the exposition in the paper it is straightforward to infer how the proposed method differs from previous work, but this is never explicitly discussed. I would suggest that the authors rework the discussion of related work (see suggestions above) and include explicit comparisons of previous work with their proposed method. Also, as this paper examines the problem of causal discovery, I would suggest the authors include related work based on the principle of independence of cause and mechanism.

Reproducibility: Yes

Additional Feedback: The claim in line 33/34 “discover the causal graph with the fewest parameters” does not necessarily apply to all score-based methods. Please quantify when it applies/. I would be happy to adjust my score upwards if the discussion of related work would be fixed and the proposed method EntropicPC compared with extensions of the PC algorithm (not just plain PC).


Review 4

Summary and Contributions: This paper explores the usefulness of common entropy for causal discovery purposes in small graphs of up to 3 variables. The authors contribute the first application of common entropy to any task in the field of causal discovery. In particular, the paper includes 3 main contributions: 1) definition of Renyi common entropy and an algorithm for the estimation of it in practice 2) the use of two special cases thereof for the purpose of distinguishing two fundamental types of causal graphs 3) the use of the distinguishing procedure above for the enhancement of the PC constraint-based causal discovery algorithm.

Strengths: The paper proposes an original contribution to the field of causal discovery, by suggesting a new basic addition to the toolbox of causal discovery. While most approaches to causal discovery rely on a sequential search over nodes while applying a conditional independence constraint, or scoring based approaches, the authors originally propose to enhance some of those by using the notion of entropy from information theory, and especially the notion of common entropy. The idea is to search for a possible latent confounder to describe a correlation between two observed variables via thresholding based on the notion of Renyi common entropy. The authors thus propose an interesting, novel and potentially useful idea that could become relevant across various applications in causal inference.

Weaknesses: The paper is trying to cover a fair amount of ground, and as such seems to glance over important details, and does not substantiate its claim with an equal amount of attention to all parts. Here are my main concerns: 1) Figure 1 considers 3 scenarios of interests to describe a causal graph for a distribution p(x,y) where x and y are correlated. However, the paper proceeds to consider another scenario, where X and Y are *mediated* by a latent, which we didn't consider in the introduction, and the significance of which is not fully discussed. Furthermore, the triangle graph is often discussed in contrast to the latent graph, but the direct graph is often forgotten. For certain applications, the difference between the direct graph and triangle graph might make a big difference (when interested in direct effects, fairness concerns, etc.). The paper seems to slip constantly between grouping triangle/direct graphs or simply dropping direct graph all together (see lines 156-157, algorithm 2, corollary 2 vs. 149, Theorem 2, 162-163). In fact, assumption 2 seems to relate to direct graphs and not triangles, but used for some reason in algorithm 2 to describe triangle graphs. I might be missing something, but I believe more care is needed in distinguishing these cases and when/why a distinction is made or not. If triangle and direct graphs are indeed interchangeable in certain cases, consider using a global term to describe both to clarify things. 2) Section 2 and 3 move back and forth between Rnenyi_0 and Renyi_1 common entropies, without much justification. While Theorem 2 is clearly established for Rejnyi_0, Assumptions 1 and 2 are stated as a general case for Renyi entropy. I am not sure why that is a fair statement (especially as Assumption 1 is never justified elsewhere and Assumption 2 is established via empirical experiments). Theorem 3 and Conjecture 1 are stated in terms of Renyi_1, and in practice, Algorithm 2 and 3, which are the main contributions, seem to be stated in terms of Renyi_1. Why include Renyi_0 at all then? Am I missing something? Perhaps clarifications or smoother movements should be included between the two? Furthermore, most readers are likely to be familiar with Shannon Entropy, so why not make the connection between Renyi_1 and Shannon Entropy more explicit? 3) The method seems to rely on tuning various parameters (\beta for the loss function (2), T as dependency threshold in Algorithm 2, \alpha in Conjecture 1, which is hardcoded to 0.8 in Algorithm 3) while searching over candidate *latent* variables. Moreover, Assumption 1 and 2 seem to be the key to much of what this paper proposes, and are mostly established via empirical results. That seems problematic and casts some doubts as to the general usability of the method. While I still think it is a very interesting proposal, clear indication of limitations, and explicit caution needs to be stated clearly. For a more detailed list of my comments and questions, see additional feedback section below.

Correctness: To the best of my knowledge and understanding, the proposed methodology seems correct, including its expansions and derivations in the appendix. However, occasional inaccuracies in presentation and writing makes correctness more obscure at times (see comments and suggestions for authors below)

Clarity: The paper could use rewrites in various places, and while its main ideas are pretty much clear, organizational as well as explanatory choices could be improved on in my opinion. Please refer to my comments and question in the additional feedback section below.

Relation to Prior Work: Common entropy has also been established elsewhere, notably in https://ieeexplore.ieee.org/stamp/stamp.jsp?tp=&arnumber=6874815, but without a connection to causal inference or discovery. The use of common entropy for causal inference and causal discovery seems novel -- I believe it differs substantially from previous contributions in the field. Previous works in causal discovery, while some use information theoretic notions, did not consider common entropy before for the identification of latent variables to the best of my knowledge.

Reproducibility: Yes

Additional Feedback: Abstract ======= 1) The abstract claims the paper “propose[s] a modification to these constraint-based causal discovery methods”. In practice, it only deals with the PC algorithm. While the authors do claim the method can be easily extended FCI and others as well, the statement in the abstract is misleading. Section 1 ======= 2) The paper is concerned with a search over latent variables. While the text refers to such a latent as Z, much of literature on causal inference would use U to denote latent variables. For consistency’s sake, consider changing Z to U. 3) More on notation: G is used for half of the paper to refer to common entropy, but refers to Graph in the other half. Notice how in algorithm 3 for example, G refers exclusively to the graph and not to common entropy anymore. Why not choose a different character for common entropy to begin with? 4) Figure 1 can be drawn to be more helpful, including the cases and distinctions this paper actually cares about (see weaknesses section). 5) Consider adding q to notation sub-section, given how often it is used later. 6) Line 27-8: Learning a causal graph is not the first step for any causal inference task. Perhaps moderate this statement? 7) Given lines 53-58, one might expect we want the latent not simply be simple, but how simple it is should be relative to the covariates X and Y in question. Perhaps clarify? (this repeats in section 3 as well). Section 2 ======= 8) Section 2, line 129: qi(z|x) appears twice. I assume the second should be qi(z|y)? 9) Section 2, line 128-129: This line is very helpful. Why is this step not shown explicitly in Algorithm 1? 10) Algorithm 1: why is beta >*=* 0? Can we actually allow \beta = 0? Wouldn’t that mean we are not enforcing anything about the entropy of the latent anymore? 11) line 131-2 is helpful in understanding where Algorithm 1 comes from. Consider expanding on the derivation in the appendix and explicitly leading the readers to the relevant part in the appendix? 12) Line 138: for \beta <= 1? Or bounded away from 0 and 1? Perhaps make the interval more explicit for clarity? Section 3 ======= 13) While the section is distinguishing between only spurious association and some level of causation, the triangle graph still includes some spurious association, i.e. it does not cleanly separate causation from spurious correlation. Consider changing the section title to be more faithful to this fact. 14) Line 157 states the common entropy of the observed variables is “large” for almost all distributions coming from the triangle graph. Line 168 explained that according to assumption 1, a latent confounder has to be “simple”. These adjectives seem too global -- don’t these “size” or “complexity” statements relative to the entropy of the observed variables? Perhaps clarify that these indications are relative to a specific system in question? 15) Line 182 refers to section 7.6. That is the appendix. Perhaps change that to link directly to the appendix, and make it clearer? Section 4 ======= 16) Line 198: typo, “fo” at the end of the line should be “to” 17) Lines 202-205 point out the weaknesses in existing constraint based causal discovery algorithms, but do not clearly state (and neither is it stated later on, as far as I can tell), that entoptic enhancement solves all these concerns. 18) Algorithm 3 uses overcomplicated and crowded notation compared to other presentations of the PC algorithm (see Spirtes et al 2001, section 6.2, or https://arxiv.org/pdf/1502.02454.pdf#:~:text=The%20PC%20algorithm%20is%20the,data%2C%20e.g.%20gene%20expression%20datasets. , algorithm 1). Consider simplifying accordingly, given that only lines 10, 13, 15 seem like the modifications. 19) Hardcoding 0.8 in Algorithm 3 and line 217 doesn’t seem properly explained. 20) Line 222, line 10 is mentioned for the second time and is bolded. Consider dropping, I think it’s clear enough given line 220. Section 5 ======= 21) the description in section 5.1 makes it sound like bigger n values were used unsuccessfully. Is that the case? If so, consider giving full details on what worked and what didn’t. 22) Why is assumption 1 never tested empirically like assumption 2? 23) The figures are out of order and do not follow the text order. That is unnecessarily confusing. 24) Figure 2a: Having y axes differ makes it hard to assess at first sight. Furthermore, left: why would we want the recovered entropy to be smaller than real? Don't we want to see some measure of loss (i.e. via loss function you defined?) 25) Why do Figure 2b and c seem to indicate worse performance for smaller n? I’m probably misinterpreting it, but perhaps clarification would be helpful. 26) Figure 3(c): the choice of Y axis seems arbitrary and not explained. Why is it not comparing directly to assumption 2? And where did the \hat{M} notation come from? 27) Section 5.5: Entropic-C seems hardly helpful given the performance reported in Figure 4. Is that true? If so, why include it? In what cases is it helpful? If not, where am I reading it wrong? 28) The synthetic data experiments were carried out on samples of size 1000, 5000 and 80000. Is the last one a typo? Should it say 8k? And if not, why choose these unexplained jumps between sample sizes? 29) Line 272 claims a “very significant” improvement. I don’t know how to base levels of significance in this case, this seems like a qualitative statement. Consider qualifying this statement? Appendix ======= 30) Like the Figures, the appendix is all out of order. E.g. Why is Theorem 1 in section 7.8-9? Please reorder the appendix chronologically + consider referring to relevant sections in the appendix from the main text?

[Author Response · NeurIPS 2020]

We would like to thank all the reviewers for their time. All reviewers acknowledged the novelty of our paper. R1 is
mainly concerned with the lack of finite-sample theory. R2 seems to have misunderstood our motivation. R3 requested
additional experiments. We address these and the other questions below. All minor suggestions will be implemented.

**R1:** *The main weakness is the missing contribution.* The idea of utilizing common entropy for causality is new. The
algorithmic approach for approximating common entropy is also new. The proposal of using all these together to
improve the constraint-based methods is also novel. We agree it would be desirable to complement our results with
finite-sample theory. Note that for this, at the very least we need good analytical bounds on common entropy, which are
currently unknown. We believe our algorithmic contribution complemented with the experimental evidence will open
up future research and lead to a better understanding of this fundamental quantity. To clarify this, we will add a *Future*
*Directions* section that lays out these next steps. We kindly ask you to reevaluate your score in light of this.

*References and discussions.* We will add the citation to the reference you suggested and also add a detailed discussion
on [23], which was dropped due to space constraints. Please see: "In [23], authors identify an assumption on $p(Y|X)$
which implies that there does not exist any latent variable $Z$ with small support which can make $X, Y$ conditionally
independent. For discrete variables, this assumption implies that the conditionals $p(Y|x)$ lie on the boundary of the
probability simplex, which corresponds to the joint probability matrix to be sparse in a structured way." Even though
the high level idea here is similar, ours is a completely different approach.

**R2:** Thank you for your feedback. We believe that you may have misunderstood our motivation and contributions. We
kindly ask you to reevaluate your assessment in light of our responses below.

*the motivation ... is to avoid the issue of high dimensional covariate.* This is not true. The motivation is to understand if
the fundamental information-theoretic quantity of common entropy can help improve/complement the existing causal
discovery methods. We provide theory and experiments that answers this question in the positive.

*ADULT data has no ground truth.* Causal inference suffers from this problem. Please see synthetic experiments instead.

*Prior work not there.* We do cite all of the related work that we are aware of. We will add the one suggested by R1 and
the discussion around it. Please recommend the article that you think is missing.

**R3:** *Experimental evaluation and comparing w/ extensions of PC.* This is a good suggestion and we can definitely
implement. Note that our goal was to provide evidence to performance improvement due only to our method without
conflating with other improvements on the PC algorithm. We believe comparing with baseline PC was necessary to
showcase the performance improvement. Modifying any extension of PC is trivial and since the reviewer thinks this is
valuable, we will add these in camera-ready.

*Minor suggestions about writing.* These are very valuable and we will make sure to implement all for clarity.

*Related work discussions.* This has been an issue due to space constraints. We will make sure to include all the extended
discussions we had to drop for space in the camera-ready. Also, please see the answer to R1. Thank you for your time.

**R5:** *Direct graph discussions.* This is a great point and for completeness, we will provide more details. Note that proofs
for distinguishing direct graph from the latent graph are "easier" than the proofs for distinguishing triangle graph from
the latent graph. This is because triangle graph brings in more parameters which should be handled carefully (see Proof
of Thm. 2) This is the main reason we focused on triangle/latent distinction. For completeness, we will complement the
writing by including discussions around the direct graph and the associated proofs in the camera-ready.

*Renyi 1, Renyi 0, Shannon entropy* We will add the statement that Renyi 1 and Shannon entropies are identical. Our
goal for using Renyi entropy is to unify different notions of complexity of the latent variable. In some cases, it might
be reasonable to assume the support size of the confounder is small. In others, we might have to relax this and
Renyi 1/Shannon entropy is a way to do that. From a fundamental point of view, understanding which Renyi entropy
assumptions lead to identifiability seem important and although we focused on two special cases, in the future we hope
to gain a better understanding for the whole spectrum of Renyi entropies and flesh out their connections to identifiability.

*Clarifying limitations/assumptions*: Thank you. We will further emphasize the limitations of the method, and the fact
that it relies on the assumptions provided.

*Citing Kumar et al.* This publication has indeed been our starting point, as we cite in line 73 in pg. 2.

*Detailed comments.* We will use all the suggestions to improve writing/clarity. We are confident this should be doable
with the extra page provided in the camera-ready. Thank you.

*Testing Assumption 1 on real data.* As far as we know, there is simply no real data that is known to fit latent graph,
whereas Tubingen provides one for causally related variables.

*80k samples.* This is a sanity check and is used as a proxy for infinite samples. Performances coincide as expected.

[Meta-Review · NeurIPS 2020]

While the reviewers took issue with the similarity to prior work, the lack of theoretical evidence around the small-sample improvement claim, the lack of experimental comparison with related work, and clarity of the presentation, I believe the authors answered these concerns sufficiently in their rebuttal, and the work is novel methodologically and theoretically to warrant acceptance. First, the reviewers had the following main concerns: - Concern #1 (R1): Idea is too similar to [23]. > Author response: In [23] the authors make an assumption which allows them to rule out the existence of an unobserved confounder Z that makes cause X and effect Y independent. They do so by assuming Z is 'low-complexity' which means either discrete or compact (in their experiments they only consider discrete Z). This is different from our approach. -> My take: I agree with the authors. The assumption here (Assumption 1 in the paper) is that any unobserved confounder Z has low entropy, with known upper bound. Given this, they can distinguish between spurious correlation between cause X and effect Y, and X directly causing Y. This is a different take on 'low-complexity', using entropy. While the intuition is the same (confounder is simple in some respect) the assumptions on the confounder are totally different. - Concern #2 (R1): The authors claim their method is beneficial in the small-sample regime but there is no theoretical evidence. > Author response: Finite sample theory for entropy is very difficult, currently good analytical bounds on common entropy are unknown, and would be necessary. -> My take: The authors already show experimentally that their method is more stable as sample size is reduced (from 80,000 to 1,000) than the standard PC algorithm and improves upon it everywhere. This makes sense as one is making stronger assumptions than the standard PC algorithm (i.e., a known upper bound on the entropy of all unobserved confounders), but it is very interesting to see what those additional assumptions buy you. - Concern #3 (R2): The motivation of the method is to avoid a high dimensional X and Y. Further the evaluation on Adult is poor because they assume a causal ground truth when there isn't one. > Author response: This is not our motivation, the motivation is to understand if making an assumption on the maximum allowed entropy of a confounder can lead to improved causal discovery. Further, we use a common way to evaluate causal discovery methods, by using data to fit the parameters of a fixed causal graph. -> My take: I completely agree with the authors here. - Concern #4 (R3): The authors do not compare to related work such as extensions of the PC algorithm. > Author response: Our goal was to isolate improvements given by our method from those given by extensions to PC, we will include this additional comparison. -> My take: Extensions to the PC algorithm are completely orthogonal to this work. In fact the choice of the PC algorithm is arbitrary, I believe this approach in this work could have been used to modify other causal discovery methods such as FCI. This suggested comparison is interesting, as there may be assumptions in PC extensions that have an overlapping effect with the current approach, but this approach can be applied to any causal discovery method, potentially yielding added benefits. - Concern #5 (R5): There are a number of presentation issues: (a) The introduction of the paper seems to mislead readers by focussing on distinguishing correlation between X and Y and an actual causal effect between X and Y, whereas the rest of the paper considers a mediation scenario where the causal effect between X and Y is mediated by Z. The significance of the mediation graph needs to be better discussed. (b) Direct graph (X->Y) is often forgotten in discussions (c) Authors seem to arbitrarily switch between different types of entropies without explanation (d) Better discussion of tuning parameters and limitations > Author response: They will make (b), (c), (d) clearer. -> My take: Based on the content of the paper and the rebuttal I believe they will be able to improve paper clarity. I urge the authors to also fix point (a). So I think the authors adequately addressed all of the reviewers concerns. Beyond these concerns I believe the paper warrants acceptance for the following reasons: - The idea of using a maximum entropy assumption on unobserved confounders to improve cause-effect identifiability is novel to my knowledge. While this isn't the first work to make some assumption on unobserved confounders to improve identifiability, the type of assumption is new. - They introduce a novel relaxation for computing the Renyi entropy of two random variables, and an algorithm for discovering the joint distribution q(x,y,z). I encourage the authors to include a proof of Theorem 1 either in the main paper or the supplement. - They have insightful experiments on: the performance of LatentSearch (its error for different dimensionalities of X,Y,Z), a test of a conjecture about how Renyi-1 entropy is bounded by the minimum of the entropies of H and Y, procedures on how to set \alpha based on a cause-effect dataset, and an evaluation of algorithms 2 and 3. I am impressed by the amount of analysis here. I think the biggest issue with the paper is presentation. There are many details to cover here. I would suggest the authors to more carefully and comprehensively describe sections 1,3,4,5 and potentially move all of section 2 (except the definition of Renyi entropy and other details that make the following sections clear) to the supplement. While Algorithm 1 and the relaxation are novel, I think it detracts from the core of the paper: using an assumption on the entropy of the latent confounder to improve identifiability. Ultimately how Renyi entropy is calculated seems somewhat orthogonal to the rest of the paper, so long as there is discussion/experiments on the behaviour of the method and how that behaviour impacts the identifiability results (I think the experiments already do this well). Also, I urge the authors to improve the presentation of the paper based on the clarity suggestions of the reviewers.